# No-Regret Learning for Fair Multi-Agent Social Welfare Optimization

**Mengxiao Zhang**
University of Iowa
mengxiao-zhang@uiowa.edu

**Ramiro Deo-Campo Vuong**
Cornell University
ramdcv@cs.cornell.edu

**Haipeng Luo**
University of Southern California
haipengl@usc.edu

## Abstract

We consider the problem of online multi-agent Nash social welfare (NSW) maximization. While previous works of Hossain et al. [2021], Jones et al. [2023] study similar problems in stochastic multi-agent multi-armed bandits and show that $\sqrt{T}$-regret is possible after $T$ rounds, their fairness measure is the product of all agents' rewards, instead of their NSW (that is, their geometric mean). Given the fundamental role of NSW in the fairness literature, it is more than natural to ask whether no-regret fair learning with NSW as the objective is possible. In this work, we provide a complete answer to this question in various settings. Specifically, in stochastic $N$-agent $K$-armed bandits, we develop an algorithm with $\widetilde{\mathcal{O}}(K^{\frac{2}{N}}T^{\frac{N-1}{N}})$ regret and prove that the dependence on $T$ is tight, making it a sharp contrast to the $\sqrt{T}$-regret bounds of Hossain et al. [2021], Jones et al. [2023]. We then consider a more challenging version of the problem with adversarial rewards. Somewhat surprisingly, despite NSW being a concave function, we prove that no algorithm can achieve sublinear regret. To circumvent such negative results, we further consider a setting with full-information feedback and design two algorithms with $\sqrt{T}$-regret: the first one has no dependence on $N$ at all and is applicable to not just NSW but a broad class of welfare functions, while the second one has better dependence on $K$ and is preferable when $N$ is small. Finally, we also show that logarithmic regret is possible whenever there exists one agent who is indifferent about different arms.

## 1 Introduction

In this paper, we study online multi-agent Nash social welfare (NSW) maximization, which is a generalization of the classic multi-armed bandit (MAB) problem [Thompson, 1933, Lai and Robbins, 1985]. Different from MAB, in which the learner makes her decisions sequentially in order to maximize her own reward, in online multi-agent NSW maximization, the learner's decision affects multiple agents and the goal is to maximize the NSW over all the agents. Specifically, NSW is defined as the geometric mean of the expected utilities over all agents [Moulin, 2004], which can be viewed as a measure of fairness among the agents. This problem includes many important real-life applications such as resource allocation [Jones et al., 2023], where the learner needs to guarantee fair allocations among multiple agents. We refer the readers to [Hossain et al., 2021, Jones et al., 2023] for more applications of NSW maximization.

Recent work by Hossain et al. [2021], Jones et al. [2023] studies a similar problem but with $\text{NSW}_{\text{prod}}$ as the objective, a variant of NSW that is defined as the product of the utilities over agents instead of

their geometric mean. While the optimal strategy is the same if the utility for each agent is stationary, this is not the case with a non-stationary environment. Moreover, $\text{NSW}_{\text{prod}}$ is homogeneous of degree $N$ instead of degree 1, where $N$ is the number of agents, meaning that $\text{NSW}_{\text{prod}}$ is more sensitive to the scale of the utility. Specifically, if the utilities of each agent are scaled by 2, then NSW is scaled by 2 as well, but $\text{NSW}_{\text{prod}}$ is scaled by $2^N$. Therefore, it is arguably more reasonable to consider regret with respect to NSW, which has not been studied before (to our knowledge) and is the main objective of our work.

From a technical perspective, however, due to the lack of Lipschitzness, NSW poses much more challenges in regret minimization compared to $\text{NSW}_{\text{prod}}$. For example, one cannot directly apply the algorithm for Lipschitz bandits [Kleinberg et al., 2019] to our problem, while it is directly applicable to $\text{NSW}_{\text{prod}}$ as mentioned in [Hossain et al., 2021, Jones et al., 2023]. Despite such challenges, we manage to provide complete answers to this problem in various setting. Specifically, our contributions are listed below (where $T, N$, and $K$ denote the number of rounds, agents, and arms/actions respectively):

- (Section 3) We first study the stochastic bandit setting, where the utility matrix at each round is i.i.d. drawn from an unknown distribution, and the learner can only observe the utilities (for different agents) of the action she picked. In this case, we develop an algorithm with $\widetilde{\mathcal{O}}(K^{\frac{2}{N}} T^{\frac{N-1}{N}})$ regret.[1] While our algorithm is also naturally based on the Upper Confidence Bound (UCB) algorithm as in Hossain et al. [2021], Jones et al. [2023], we show that a novel analysis with Bernstein-type confidence intervals is important for handling the lack of Lipschitzness of NSW. Moreover, we prove a lower bound of order $\widetilde{\Omega}(\frac{1}{N^3} \cdot K^{\frac{1}{N}} T^{\frac{N-1}{N}})$, showing that the dependence on $T$ is tight. This is in sharp contrast to the $\sqrt{T}$-regret bound of Hossain et al. [2021], Jones et al. [2023] and demonstrates the difficulty of learning with NSW compared to $\text{NSW}_{\text{prod}}$.

- (Section 4.1) We then consider a more challenging setting where the utility matrix at each round can be adversarially chosen by the environment. Somewhat surprisingly, we show that no algorithm can achieve sublinear regret in this case, despite NSW being concave and the vast literature on bandit online maximization with concave utility functions (the subtlety lies in the slightly different feedback model). In fact, the same impossibility result also holds for $\text{NSW}_{\text{prod}}$ as we show.

- (Section 4.2) To bypass such impossibility, we further consider this adversarial setting under richer feedback, where the learner observes the full utility matrix after her decision (the so-called full-information feedback). Contrary to the bandit feedback setting, learning is not only possible now but can also be much faster despite having adversarial utilities. Specifically, we design two different algorithms with $\sqrt{T}$-regret. The first algorithm is based on Follow-the-Regularized-Leader (FTRL) with the log-barrier regularizer, which achieves $\mathcal{O}(\sqrt{KT \log T})$ regret (Section 4.2.1). Notably, this algorithm does not have any dependence on the number of agents $N$ and can also be generalized to a broader class of social welfare functions. The second algorithm is based on FTRL with a Tsallis entropy regularizer, which achieves $\widetilde{\mathcal{O}}(K^{\frac{1}{2} - \frac{1}{N}} \sqrt{NT})$ regret and is thus more favorable when $K$ is much larger than $N$ (Section 4.2.2). Finally, we also show that improved logarithmic regret is possible as long as at each round there exists at least one agent who is indifferent about the learner's choice of arm (Section 4.2.3).

## 1.1 Related Work

Hossain et al. [2021], Jones et al. [2023] are most related to our work. Hossain et al. [2021] is the first to consider designing no-regret algorithms under $\text{NSW}_{\text{prod}}$ for the stochastic multi-agent multi-armed bandit problem. Specifically, they propose two algorithms. The first one is based on $\varepsilon$-greedy and achieves $\mathcal{O}(T^{\frac{2}{3}})$ regret efficiently, and the second one is based on UCB and achieves $\widetilde{\mathcal{O}}(\sqrt{T})$ regret inefficiently. Jones et al. [2023] improves these results by providing a better UCB-based algorithm that is efficient and achieves the same $\widetilde{\mathcal{O}}(\sqrt{T})$ regret. To the best of our knowledge, there are no previous results for regret minimization over NSW under this particular setup.

However, several other models of fairness have been introduced in (single-agent or multi-agent) multi-armed bandits, some using NSW as well. These models differ in whether they aim to be fair among different objectives, different arms, different agents, different rounds, or others. Most

---

[1]The notation $\widetilde{\mathcal{O}}(\cdot)$ and $\widetilde{\Omega}(\cdot)$ hide logarithmic dependence on $K$, $N$, and $T$.

related to this paper is multi-objective bandits, in which the learner tries to increase different and possibly competing objectives in a fair manner. For example, Drugan and Nowe [2013] introduces the multi-objective stochastic bandit problem and offers a regret measure to explore Pareto Optimal solutions, and Busa-Fekete et al. [2017] investigates the same setting using the Generalized Gini Index in their regret measure to promote fairness over objectives. Their regret measure closely resembles the one we use, except they apply some social welfare function (SWF) to the cumulative expected utility of agents over all rounds as opposed to the expected utility of agents each round. On the other hand, some other works study fairness among different rounds which incentivizes the learner to perform well consistently over all rounds [Barman et al., 2023, Sawarni et al., 2024]. Besides, there are other models that measure fairness in different ways, including how often each arm is pulled [Joseph et al., 2016, Liu et al., 2017, Gillen et al., 2018, Chen et al., 2020] and how the regret is allocated across different groups [Baek and Farias, 2021].

Kaneko and Nakamura [1979] axiomatically derives the NSW function. It is a fundamental and widely-adopted fairness measure and is especially popular for the task of fairly allocating goods. Caragiannis et al. [2019] justifies the fairness of NSW by showing that its maximum solution ensures some desirable envy-free property. This result prompted the design of approximation algorithms for the problem of allocating indivisible goods by maximizing NSW, which is known to be NP-hard even for simple valuation functions [Barman et al., 2018, Cole and Gkatzelis, 2015, Garg et al., 2023, Li and Vondrák, 2021].

There is a vast literature on the multi-armed bandit problem; see the book by Lattimore and Szepesvári [2020] for extensive discussions. The standard algorithm for the stochastic setting is UCB [Lai and Robbins, 1985, Auer et al., 2002a], while the standard algorithm for the adversarial setting is FTRL or the closely related Online Mirror Descent (OMD) algorithm [Auer et al., 2002b, Audibert and Bubeck, 2010, Abernethy et al., 2015]. For FTRL/OMD, the log-barrier or Tsallis entropy regularizers have been extensively studied in recent years due to some of their surprising properties (e.g., [Foster et al., 2016, Wei and Luo, 2018, Zimmert and Seldin, 2019, Lee et al., 2020]). They are rarely used in the full-information setting as far as we know, but our analysis reveals that they are useful even in such settings, especially for dealing with the lack of Lipschitzness of NSW.

## 2 Preliminaries

**General Notation.**    Throughout this paper, we denote the set $\{1, 2, \ldots, n\}$ by $[n]$ for any positive integer $n$. For a matrix $M \in \mathbb{R}^{m \times n}$, we denote the $i$-th row vector of $M$ by $M_{i,:} \in \mathbb{R}^n$, the $j$-th column vector of $M$ by $M_{:,j} \in \mathbb{R}^m$, and the $(i, j)$-th entry of $M$ by $M_{i,j}$. We say $M \succeq 0$ if $M$ is a positive semi-definite matrix. The $(K - 1)$-dimensional simplex is denoted as $\Delta_K$, and its clipped version with a parameter $\delta > 0$ is denoted as $\Delta_{K,\delta} = \{p \in \Delta_K \mid p_i \geq \delta, \forall i \in [K]\}$. We use $\mathbf{0}$ and $\mathbf{1}$ to denote the all-zero and all-one vector in an appropriate dimension. For two random variables $X$ and $Y$, we use $X \overset{d}{=} Y$ to say $X$ is equivalent to $Y$ in distribution.

For a twice differentiable function $f$, we use $\nabla f(\cdot)$ and $\nabla^2 f(\cdot)$ to denote its gradient and Hessian. For concave functions that are not differentiable, $\nabla f(\cdot)$ denotes a super-gradient. Throughout the paper, we study functions of the form $f(u^\top p)$ for $u \in [0, 1]^{m \times n}$ and $p \in \Delta_m$. In such cases, the gradient, super-gradient, or hessian are all with respect to $p$ unless denoted otherwise (for example, we write $\nabla_u f(u^\top p)$, with an explicit subscript $u$, to denote the gradient with respect to $u$).

**Social Welfare Functions**    A social welfare function (SWF) $f : [0, 1]^N \to [0, 1]$ measures the desirability of the agents' expected utilities. Specifically, for two different vectors of expected utilities $\mu, \mu' \in [0, 1]^N$, $f(\mu) > f(\mu')$ means that $\mu$ is a fairer alternative than $\mu'$. In each setting we explore, each action by the learner yields some expected utility for each of the $N$ agents, and the learner's goal is maximize some SWF applied to these $N$ expected utilities.

**Nash Social Welfare (NSW)**    For the majority of this paper, we focus on a specific type of SWF, namely the Nash Social Welfare (NSW) function [Nash, 1950, Kaneko and Nakamura, 1979]. Specifically, for $\mu \in [0, 1]^N$, NSW is defined as the geometric mean of the $N$ coordinates:

$$\text{NSW}(\mu) = \prod_{n \in [N]} \mu_n^{1/N}. \tag{1}$$

As mentioned, Hossain et al. [2021], Jones et al. [2023] considered a closely related variant that is simply the product of the coordinates: $\text{NSW}_{\text{prod}}(\mu) = \prod_{n \in [N]} \mu_n$. It is clear that NSW has a better scaling property since it is homogeneous: scaling each $\mu_n$ by a constant $c$ also scales $\text{NSW}(\mu)$ by $c$, but it scales $\text{NSW}_{\text{prod}}(\mu)$ by $c^N$. This makes $\text{NSW}_{\text{prod}}$ an unnatural learning objective, which motivates us to use NSW as our choice of SWF. Learning with NSW, however, brings extra challenges since it is not Lipschitz in the small-utility regime (while $\text{NSW}_{\text{prod}}$ is Lipschitz over the entire $[0,1]^N$). We shall see in subsequent sections how we address such challenges.

We remark that while our main focus is regret minimization with respect to NSW, some of our results also apply to $\text{NSW}_{\text{prod}}$ or more general classes of SWFs (as will become clear later).

**Problem Setup.** The $N$-agent $K$-armed social welfare optimization problem we consider is defined as follows (with $N \geq 2$ and $K \geq 2$ throughout). Ahead of time, with the knowledge of the learner's algorithm, the environment decides $T$ utility matrices $u_1, \ldots, u_T \in [0,1]^{K \times N}$, where $u_{t,i,n}$ is the utility of agent $n$ if arm/action $i$ is selected at round $t$. Then, the learner interacts with the environment for $T$ rounds: at each round $t$, the learner decides a distribution $p_t \in \Delta_K$ and then samples an action $i_t \sim p_t$. In the full-information feedback setting, the learner observes the full utility matrix $u_t$ after her decision, and in the bandit feedback setting, the learner only observes $u_{t,i_t,n}$ for each agent $n \in [N]$, that is, the utilities of the selected action.

We consider two different types of environments, the *stochastic* one and the *adversarial* one, with a slight difference in their regret definition. In the stochastic environment, there exists a mean utility matrix $u \in [0,1]^{K \times N}$ such that at each round $t$, $u_t$ is an i.i.d. random variable with mean $u$. Fix an SWF $f$. The social welfare of a strategy $p \in \Delta_K$ is defined as $f(u^\top p)$, which is with respect to the agents' expected utilities over the randomness of both the learner's and the environment's. The regret is then defined as follows:

$$\text{Reg}_{\text{sto}} = T \cdot \max_{p \in \Delta_K} f(u^\top p) - \mathbb{E}\left[\sum_{t=1}^{T} f(u^\top p_t)\right], \tag{2}$$

which is the difference between the total social welfare of the optimal strategy and that of the learner. When $f$ is chosen to be $\text{NSW}_{\text{prod}}$, Eq. (2) reduces to the regret notion considered in Hossain et al. [2021], Jones et al. [2023].

On the other hand, in the adversarial environment, we do not make any distributional assumption on the utility matrices and allow them to be selected arbitrarily. The social welfare of a strategy $p \in \Delta_K$ for time $t$ is defined as $f(u_t^\top p)$, and the overall regret of the learner is correspondingly defined as:

$$\text{Reg}_{\text{adv}} = \max_{p \in \Delta_K} \sum_{t=1}^{T} f(u_t^\top p) - \mathbb{E}\left[\sum_{t=1}^{T} f(u_t^\top p_t)\right]. \tag{3}$$

In both Eq. (2) and Eq. (3), the expectation is taken with respect to the randomness of the algorithm.

**Social welfare of expected utilities versus expected social welfare of realized utilities.** One might wonder why we measure fairness using the social welfare of expected utilities (e.g., $f(u^\top p)$), instead of the expected social welfare of realized utilities (e.g., $\mathbb{E}_{i \sim p}[f(u^\top e_i)]$). This is because the former is arguably more meaningful as a fairness measure. To see this, consider $f = \text{NSW}$ or $f = \text{NSW}_{\text{prod}}$ and a setting with 2 agents, 2 arms, and $u$ being the identity matrix. Then, in terms of $f(u^\top p)$, the uniform distribution is the best policy (which makes sense from a fairness viewpoint), while in terms of $\mathbb{E}_{i \sim p}[f(u^\top e_i)]$, all distributions achieve the same value of 0, implying that all polices are as fair, which is clearly undesired.

**Connection to Bandit Convex optimization.** When taking $f = \text{NSW}$ (our main focus) and considering the bandit feedback setting, our problem is seemingly an instance of the heavily-studied Bandit Convex optimization (BCO) problem, since $-\text{NSW}$ is convex. However, there is a slight but critical difference in the feedback model: a BCO algorithm would require observing $f(u_t^\top p_t)$, or equivalently $u_t^\top p_t$, at the end of each round $t$, while in our problem the learner only observes $u_{t,i_t,:}$, a much more realistic scenario. Even though they have the same expectation, due to the non-linearity of NSW, this slight difference in the feedback turns out to cause a huge difference in terms of learning — the minimax regret for BCO is known to be $\Theta(\sqrt{T})$, while in our problem (with bandit feedback),

**Algorithm 1** UCB for $N$-agent $K$-armed NSW maximization

---

Input: warm-up phase length $N_0 > 0$.
Initialization: $\widehat{u}_{1,i,n} = 1$ for all $n \in [N]$, $i \in [K]$. $N_{1,i} = 0$ for all $i \in [K]$.
**for** $t = 1, 2, \ldots, T$ **do**

    **if** $t \leq KN_0$ **then** select $i_t = \lceil \frac{t}{N_0} \rceil$ ;

    **else** calculate $p_t = \mathrm{argmax}_{p \in \Delta_K} \mathrm{NSW}(\widehat{u}_t^\top p)$ and select $i_t \sim p_t$ ;

    Observe $u_{t,i_t,n}$ for all $n \in [N]$.

    Update counters $N_{t+1,i_t} = N_{t,i_t} + 1$ and $N_{t+1,i} = N_{t,i}$ for $i \neq i_t$.

    Update upper confidence utility matrix:

$$\widehat{u}_{t+1,i,n} = \bar{u}_{t,i,n} + 4\sqrt{\frac{\bar{u}_{t,i,n} \log(NKT^2)}{N_{t+1,i}}} + \frac{8\log(NKT^2)}{N_{t+1,i}}, \tag{4}$$

    for all $n \in [N]$ and $i \in [K]$ where $\bar{u}_{t,i,n} = \frac{1}{N_{t+1,i}} \sum_{\tau \leq t} u_{\tau,i,n} \mathbb{1}\{i_\tau = i\}$.

**end**

---

as we will soon show, the regret is either $\Theta(T^{\frac{N-1}{N}})$ in the stochastic setting or even $\Omega(T)$ in the adversarial setting. Therefore, in a sense our problem is much more difficult than BCO. For more details on BCO, we refer the reader to a recent survey by Lattimore [2024].

# 3 Stochastic Environments with Bandit Feedback

In this section, we consider regret minimization over $f = \mathrm{NSW}$ with bandit feedback in the stochastic setting, where the utility matrix $u_t$ at each round $t \in [T]$ is i.i.d. drawn from a distribution with mean $u$. Again, this is the same setup as Hossain et al. [2021], Jones et al. [2023] except that $\mathrm{NSW}_{\mathrm{prod}}$ is replaced with NSW.

## 3.1 Upper Bound: a Refined Analysis of UCB with a Bernstein-Type Confidence Set

We start by describing our algorithm and its regret guarantee, followed by discussion on what the key ideas are and how the algorithm/analysis is different from previous work. Specifically, our algorithm, shown in Algorithm 1, is based on the classic UCB algorithm. It starts by picking each action for $N_0 = \widetilde{\mathcal{O}}(1)$ rounds. After this warm-up phase, at each time $t$, the algorithm picks the optimal strategy $p_t$ that maximizes the NSW with respect to some upper confidence utility matrix $\widehat{u}_t$. After sampling an action $i_t \sim p_t$, the algorithm observes the utility of each agent for action $i_t$ and then updates the upper confidence utility matrix $\widehat{u}_{t+1}$ as the empirical average utility plus a certain Bernstein-type confidence width (Eq. (4)).

The following theorem shows that Algorithm 1 guarantees $\widetilde{\mathcal{O}}(K^{\frac{2}{N}} T^{\frac{K-1}{K}})$ expected regret (with $f$, in the definition of $\mathrm{Reg}_{\mathrm{sto}}$, set to NSW; the same below unless stated otherwise). The full proof can be found in Appendix A.

**Theorem 3.1.** *With $N_0 = 1 + 18 \log KT$, Algorithm 1 guarantees $\mathbb{E}[\mathrm{Reg}_{\mathrm{sto}}] = \widetilde{\mathcal{O}}(K^{\frac{2}{N}} T^{\frac{N-1}{N}} + K)$.*

Other than replacing $\mathrm{NSW}_{\mathrm{prod}}$ with NSW, our algorithm differs from that of Jones et al. [2023] in the form of the confidence width, and the analysis sketch below explains why we need this change. Specifically, for either $f = \mathrm{NSW}_{\mathrm{prod}}$ or $f = \mathrm{NSW}$, standard analysis of UCB states that the regret is bounded by $\sum_{t=1}^{T} \left| f(\widehat{u}_t^\top p_t) - f(u^\top p_t) \right|$. When $f$ is $\mathrm{NSW}_{\mathrm{prod}}$, a Lipschitz function, Hossain et al. [2021, Lemma 3] shows

$$\left| \mathrm{NSW}_{\mathrm{prod}}(\widehat{u}_t^\top p_t) - \mathrm{NSW}_{\mathrm{prod}}(u^\top p_t) \right| \leq \sum_{n=1}^{N} \sum_{i=1}^{K} p_{t,i} \left| \widehat{u}_{t,i,n} - u_{i,n} \right|, \tag{5}$$

and the rest of the analysis follows by direct calculations. However, when $f$ is NSW, a non-Lipschitz function, we cannot expect something similar to Eq. (5) anymore. Indeed, direct calculation shows that the Lipschitz constant of $\mathrm{NSW}(u^\top p)$ with respect to $u_{:,n}$ equals to $\Theta\left( \sum_{n=1}^{N} \langle p, u_{:,n} \rangle^{-\frac{N-1}{N}} \right)$, which can be arbitrarily large when $\langle p, u_{:,n} \rangle$ is close to 0 for some $n \in [N]$ and $N \geq 2$.

To handle this issue, we require a more careful analysis. Specifically, using Freedman's inequality, we know that with a high probability,

$$\widehat{u}_{t,i,n} \in \left[ u_{i,n}, u_{i,n} + 8\sqrt{\frac{u_{i,n}\log(NKT^2)}{N_{t,i}}} + \widetilde{\mathcal{O}}\left(1/N_{t,i}\right) \right] \subseteq \left[ u_{i,n}, 2u_{i,n} + \widetilde{\mathcal{O}}\left(1/N_{t,i}\right) \right]. \quad (6)$$

With the help of Eq. (6), we consider two different cases at each round $t$. The first case is that there exists certain $n \in [N]$ such that $\langle p_t, u_{:,n} \rangle \leq \sigma$ for some $\sigma > 0$ to be chosen later. In this case, we use Eq. (6) to show

$$\left| \mathrm{NSW}(\widehat{u}_t^\top p_t) - \mathrm{NSW}(u^\top p_t) \right| \leq \mathcal{O}\left( \mathrm{NSW}(u^\top p_t) \right) + \widetilde{\mathcal{O}}\left( \left( \sum_{i=1}^{K} \frac{p_{t,i}}{N_{t,i}} \right)^{\frac{1}{N}} \right)$$

$$\leq \sigma^{\frac{1}{N}} + \widetilde{\mathcal{O}}\left( \left( \sum_{i=1}^{K} \frac{p_{t,i}}{N_{t,i}} \right)^{\frac{1}{N}} \right), \quad (7)$$

where the first inequality uses Eq. (6) and the second inequality is because $\mathrm{NSW}(u^\top p_t) \leq \langle p_t, u_n \rangle^{\frac{1}{N}}$ for any $n \in [N]$. For the second term in Eq. (7), a standard analysis shows that it is upper bounded by $\widetilde{\mathcal{O}}\left( K^{\frac{1}{N}} T^{\frac{N-1}{N}} \right)$.

Now we consider the case where $\langle p_t, u_{:,n} \rangle \geq \sigma$ for all $n \in [N]$. In this case, via a decomposition lemma (Lemma C.1), we show that

$$\left| \mathrm{NSW}(\widehat{u}_t^\top p_t) - \mathrm{NSW}(u^\top p_t) \right| \leq \sum_{n=1}^{N} \left[ \langle p_t, \widehat{u}_{t,:,n} \rangle^{\frac{1}{N}} - \langle p_t, u_{:,n} \rangle^{\frac{1}{N}} \right] = \mathcal{O}\left( \sum_{n=1}^{N} \frac{\langle p_t, \widehat{u}_{t,:,n} - u_{:,n} \rangle}{N \langle p_t, u_{:,n} \rangle^{\frac{N-1}{N}}} \right). \quad (8)$$

To bound Eq. (8), we use Eq. (6) again:

$$\frac{\langle p_t, \widehat{u}_{t,:,n} - u_{:,n} \rangle}{\langle p_t, u_{:,n} \rangle^{\frac{N-1}{N}}} \leq \mathcal{O}\left( \frac{1}{\langle p_t, u_{:,n} \rangle^{\frac{N-1}{N} - \frac{1}{2}}} \sum_{i=1}^{K} \sqrt{\frac{p_{t,i}}{N_{t,i}}} \right) \leq \mathcal{O}\left( \sigma^{\frac{1}{2} - \frac{N-1}{N}} \sum_{i=1}^{K} \sqrt{\frac{p_{t,i}}{N_{t,i}}} \right), \quad (9)$$

where the last inequality is due to the condition $\langle p_t, u_{:,n} \rangle \geq \sigma$ for all $n \in [N]$. Finally, combining Eq. (7), Eq. (8), Eq. (9), followed by direct calculations, we show that

$$\mathbb{E}\left[ \mathrm{Reg}_{\mathrm{sto}} \right] \leq \sum_{t=1}^{T} \left| \mathrm{NSW}(\widehat{u}_t^\top p_t) - \mathrm{NSW}(u^\top p_t) \right| \leq \widetilde{\mathcal{O}}\left( T\sigma^{\frac{1}{N}} + K^{\frac{1}{N}} T^{\frac{N-1}{N}} + \sigma^{\frac{1}{2} - \frac{N-1}{N}} K\sqrt{T} \right).$$

Picking the optimal choice of $\sigma$ finishes the proof.

We now highlight the importance of using a Bernstein-type confidence width in Eq. (4): if the standard Hoeffding-type confidence width is used instead, then one can only obtain $\widehat{u}_{t,i,n} - u_{i,n} \leq \mathcal{O}(\sqrt{\frac{1}{N_{t,i}}})$, and consequently, Eq. (8) can only be bounded by $\mathcal{O}\left( \sigma^{-\frac{N-1}{N}}\sqrt{KT} \right)$ after taking summation over $t \in [T]$. This eventually leads to a worse regret bound of $\widetilde{\mathcal{O}}(K^{\frac{1}{2N}} T^{\frac{2N-1}{2N}})$.

### 3.2 Lower Bound

Next, we prove an $\widetilde{\Omega}(T^{\frac{N-1}{N}})$ lower bound for this setting. This not only shows that the regret bound we achieve via Algorithm 1 is tight in $T$, but also highlights the difference and difficulty of learning with NSW compared to learning with $\mathrm{NSW}_{\mathrm{prod}}$, since in the latter case, $\Theta(\sqrt{T})$ regret is minimax optimal [Hossain et al., 2021, Jones et al., 2023].

**Theorem 3.2.** *In the bandit feedback setting, for any algorithm, there exists a stochastic environment in which the expected regret (with respect to NSW) of this algorithm is* $\Omega\left( \frac{(\log K)^3}{N^3} \cdot K^{\frac{1}{N}} T^{\frac{N-1}{N}} \right)$ *for* $N \geq \log K$ *and sufficiently large $T$.*

We defer the full proof to Appendix A.2 and discuss the hard instance used in the proof below. First, the mean utility vector $u_{:,n}$ for each agent $n \geq 2$ is a constant vector $\mathbf{1}$. This makes the problem

equivalent to a one-agent problem, but with $\langle p, u_{:,1} \rangle^{1/N}$ as the reward, instead of $\langle p, u_{:,1} \rangle$ as in standard stochastic $K$-armed bandits.

Then, for the first agent, different from the standard $K$-armed bandits, where the hardest instance is to hide one arm with a slightly better expected reward of $\frac{1}{2} + \sqrt{K/T}$ among other $K - 1$ arms with expected reward of exactly $\frac{1}{2}$,[2] we hide one arm with expected reward $K/T$ among other $K - 1$ arms with exactly 0 reward (so overall the rewards are shifted towards 0 but with a smaller gap between the best arm and the others). By standard information theory arguments, within $T$ rounds the learner cannot distinguish the best arm from the others. Therefore, the best strategy she can apply is to pick a uniform distribution over actions, suffering $\Omega((1 - K^{-\frac{1}{N}}) \cdot (K/T)^{\frac{1}{N}}) = \widetilde{\Omega}(K^{\frac{1}{N}} T^{-\frac{1}{N}})$ regret per round and leading to $\widetilde{\Omega}(K^{\frac{1}{N}} T^{\frac{N-1}{N}})$ regret overall.

## 4 Adversarial Environments

Now that we have a complete answer for the stochastic setting, we move on to consider the adversarial case where each $u_t$ is chosen arbitrarily, a multi-agent generalization of the expert problem (full-information feedback) [Freund and Schapire, 1997] and the adversarial multi-armed bandit problem (bandit feedback) [Auer et al., 2002b]. There are no prior studies on this problem, be it with $f = \text{NSW}$ or $f = \text{NSW}_{\text{prod}}$, as far as we know.

### 4.1 Impossibility Results with Bandit Feedback

We start by considering the bandit feedback setting. As mentioned in Section 2, even though NSW is a concave function, our problem is not an instance of Bandit Convex Optimization, since we can only observe $u_{t,i_t,:}$ instead of $\text{NSW}(u_t^\top p_t)$ at the end of round $t$. Somewhat surprisingly, this slight difference in the feedback in fact makes a sharp separation in learnability — while $\mathcal{O}(\sqrt{T})$ regret is achievable in BCO, we prove that $o(T)$ regret is impossible in our problem.

Before showing the theorem and its proof, we first give high level ideas on the construction of the hard environments. Specifically, we consider the environment with 2 agents, 2 arms, and binary utility matrix $u_t \in \{0, 1\}^{2 \times 2}$. Similar to the hard instance in the stochastic environment, we set $u_{t,:,2} = \mathbf{1}$, reducing the problem to a single-agent one. For the first agent, we let $u_{t,:,1}$ at each round $t$ be i.i.d. drawn from a stationary distribution over the 4 binary utility vectors $\{(0,0), (0,1), (1,0), (1,1)\}$. Then, we construct two different distributions, $\mathcal{E}$ and $\mathcal{E}'$, over these 4 binary utility vectors satisfying that: 1) the distribution of the learner's observation is identical for $\mathcal{E}$ and $\mathcal{E}'$; 2) the optimal strategy for $\mathcal{E}$ and $\mathcal{E}'$ are significantly different. The first property guarantees that no algorithm can distinguish these two environments, while the second property ensures that there is no one single strategy that can perform well in both environments. Formally, we prove the following theorem.

**Theorem 4.1.** *In the bandit feedback setting, for any algorithm, there exists an adversarial environment such that* $\mathbb{E}[\text{Reg}_{\text{adv}}] = \Omega(T)$ *for* $f = \text{NSW}$.

*Proof.* As sketched earlier, we consider two different environments with 2 agents, 2 arms, and binary utility matrices $u_t \in \{0, 1\}^{2 \times 2}$, $t \in [T]$. In both environments, we have $u_{t,:,2} = \mathbf{1}$. Next, we construct two different distributions from which $u_{t,:,1}$ is potentially drawn from, $\mathcal{E}$ and $\mathcal{E}'$, over $\{(0,0), (0,1), (1,0), (1,1)\}$. Specifically, $\mathcal{E}$ is characterized by $(q_{00}, q_{01}, q_{10}, q_{11}) = (\frac{4}{10}, \frac{2}{10}, \frac{1}{10}, \frac{3}{10})$, where $q_{xy}$ is the probability of the vector $(x, y)$ in $\mathcal{E}$; $\mathcal{E}'$ is characterized by $(q'_{00}, q'_{01}, q'_{10}, q'_{11}) = (\frac{3}{10}, \frac{3}{10}, \frac{2}{10}, \frac{2}{10})$, where $q'_{xy}$ is the probability of vector $(x, y)$ in $\mathcal{E}'$. With a slight abuse of notation, we write $u \sim \mathcal{E}$ for a matrix $u \in \{0, 1\}^{2 \times 2}$ if $u_{:,1}$ is drawn from $\mathcal{E}$ and $u_{:,2} = \mathbf{1}$; the same for $\mathcal{E}'$.

We argue that the learner's observations are equivalent in distribution in $\mathcal{E}$ and $\mathcal{E}'$, since the marginal distributions of the utility of each action are the same. Specifically,

- When action 1 is chosen, the distributions of the learner's observation in both $\mathcal{E}$ and $\mathcal{E}'$ are a Bernoulli random variable with mean $q_{10} + q_{11} = q'_{10} + q'_{11} = \frac{4}{10}$;

---

[2]One can show that $\Theta(\sqrt{T})$ regret is possible in this environment, thus not suitable for our purpose.

- When action $2$ is chosen, the distributions of the learner's observation in both $\mathcal{E}$ and $\mathcal{E}'$ are a Bernoulli random variable with mean $q_{01} + q_{11} = q'_{01} + q'_{11} = \frac{5}{10}$.

Direct calculation shows $p_\star = \operatorname{argmax}_{p \in \Delta_K} \mathbb{E}_{u \sim \mathcal{E}} \left[ \mathrm{NSW}(u^\top p) \right] = \left( \frac{q_{10}^2}{q_{01}^2 + q_{10}^2}, \frac{q_{01}^2}{q_{01}^2 + q_{10}^2} \right) = (0.2, 0.8)$ and $p'_\star = \operatorname{argmax}_{p \in \Delta_K} \mathbb{E}_{u \sim \mathcal{E}'} \left[ \mathrm{NSW}(u^\top p) \right] = \left( \frac{q_{10}'^2}{q_{10}'^2 + q_{01}'^2}, \frac{q_{01}'^2}{q_{10}'^2 + q_{01}'^2} \right) = (\frac{4}{13}, \frac{9}{13})$, which are constant apart from each other. Pick a threshold value $\theta = \frac{33}{130} \in (0.2, \frac{4}{13})$. Direct calculation shows that for a strategy $p$ with $p_1 \geq \theta$, we have $\mathbb{E}_{u \sim \mathcal{E}}[\mathrm{NSW}(u^\top p_\star) - \mathrm{NSW}(u^\top p)] \geq \Delta$ where $\Delta = \frac{1}{500}$; similarly, for a strategy $p$ with $p_1 < \theta$, we have $\mathbb{E}_{u \sim \mathcal{E}'}[\mathrm{NSW}(u^\top p_\star) - \mathrm{NSW}(u^\top p)] \geq \Delta$ as well. Now, given an algorithm, let $\alpha_{\mathcal{E}}$ be the probability that the number of rounds $p_{t,1} \geq \theta$ is larger than $\frac{T}{2}$ under environment $\mathcal{E}$, and $\bar{\alpha}_{\mathcal{E}'}$ be the probability of the complement of this event under environment $\mathcal{E}'$. We have,

$$\mathbb{E}_{\mathcal{E}}[\mathrm{Reg}_{\mathrm{adv}}] \geq \mathbb{E}_{\mathcal{E}} \left[ \sum_{t=1}^{T} \mathrm{NSW}(u_t^\top p_\star) - \sum_{t=1}^{T} \mathrm{NSW}(u_t^\top p_t) \right] \geq \frac{\alpha_{\mathcal{E}} T \Delta}{2},$$

$$\mathbb{E}_{\mathcal{E}'}[\mathrm{Reg}_{\mathrm{adv}}] \geq \mathbb{E}_{\mathcal{E}'} \left[ \sum_{t=1}^{T} \mathrm{NSW}(u_t^\top p'_\star) - \sum_{t=1}^{T} \mathrm{NSW}(u_t^\top p_t) \right] \geq \frac{\bar{\alpha}_{\mathcal{E}'} T \Delta}{2}.$$

Finally, since the feedback for the algorithm is the same in distribution in these two environments, we know $\alpha_{\mathcal{E}} + \bar{\alpha}_{\mathcal{E}'} = 1$, and thus

$$\max\{\mathbb{E}_{\mathcal{E}}[\mathrm{Reg}_{\mathrm{adv}}], \mathbb{E}_{\mathcal{E}'}[\mathrm{Reg}_{\mathrm{adv}}]\} \geq \frac{\mathbb{E}_{\mathcal{E}}[\mathrm{Reg}_{\mathrm{adv}}] + \mathbb{E}_{\mathcal{E}'}[\mathrm{Reg}_{\mathrm{adv}}]}{2} \geq \frac{(\alpha_{\mathcal{E}} + \bar{\alpha}_{\mathcal{E}'}) T \Delta}{4} = \Omega(T),$$

which finishes the proof. $\square$

In fact, by a similar but more involved construction (that actually requires using two agents in a non-trivial way), the same impossibility result also holds for $f = \mathrm{NSW}_{\mathrm{prod}}$; see Appendix B.1. We remark that non-linearity of $f$ in these results plays an important role in the hard instance construction, since otherwise, the optimal strategy for $\mathcal{E}$ and $\mathcal{E}'$ will be the same as they both induce the same marginal distributions.

### 4.2 Full-Information Feedback

To sidestep the impossibility result due to the bandit feedback, we shift our focus to the full-information feedback model, where the learner observes the entirety of the utility matrix $u_t$ at the end of round $t$. As mentioned, this corresponds to a multi-agent generalization of the well-known expert problem [Freund and Schapire, 1997]. We propose several algorithms for this setting, showing that the richer feedback not only makes learning possible but also leads to much lower regret.

#### 4.2.1 FTRL with Log-Barrier Regularizer

When $f$ is concave, our problem is in fact also an instance of the well-known Online Convex Optimization (OCO) [Zinkevich, 2003]. However, standard OCO algorithms such as Online Gradient Descent, an instance of the more general Follow-the-Regularized-Leader algorithm with a $\ell_2$ regularizer, require the utility function to also be Lipschitz and thus cannot be directly applied to learning NSW. Nevertheless, we will show that using a different regularizer that induces more stability than the $\ell_2$ regularizer can resolve this issue.

More specifically, the FTRL algorithm is shown in Algorithm 2, which predicts at time $t$ the distribution $p_t = \operatorname{argmin}_{p \in \Delta_K} \langle p, - \sum_{s=1}^{t-1} \nabla f(u_s^\top p_s) \rangle + \frac{1}{\eta} \psi(p)$ for some learning rate $\eta$ and some strongly convex regularizer $\psi$. Standard analysis shows that the regret of FTRL contains two terms: the regularization penalty term that is of order $1/\eta$ and the stability term that is of order $\eta \sum_t \|\nabla f(u_t^\top p_t)\|_{\nabla^{-2} \psi(p_t)}^2$ where we use the notation $\|a\|_M = \sqrt{a^\top M a}$. To deal the lack the Lipschitzness, that is, the potentially large $\nabla f(u_t^\top p_t)$, we need to find a regularizer $\psi$ so that the induced local norm $\|\nabla f(u_t^\top p_t)\|_{\nabla^{-2} \psi(p_t)}$ is always reasonably small despite $\nabla f(u_t^\top p_t)$ being large (in $\ell_2$ norm for example).

---

**Algorithm 2** FTRL for $N$-agent $K$-armed SWF maximization with full-info feedback

---

Inputs: a SWF $f$, a learning rate $\eta > 0$, and a strongly convex regularizer $\psi : \Delta_K \to \mathbb{R}$.
**for** $t = 1, 2, \ldots, T$ **do**
    Play $p_t = \operatorname{argmin}_{p \in \Delta_K} \langle p, -\sum_{s=1}^{t-1} \nabla f(u_s^\top p_s) \rangle + \frac{1}{\eta} \psi(p)$.
    Observe $u_t$.
**end**

---

It turns out that the log-barrier regularizer, $\psi(p) = -\sum_{i=1}^{K} \log p_i$, ensures such a property. In fact, it induces a small local norm not just for NSW, but also for a broad family of SWFs as long as they are concave and *Pareto optimal* — an SWF $f : [0,1]^N \to [0,1]$ is Pareto optimal if for two utility vectors $x$ and $y$ such that $x_n \geq y_n$ for all $i \in [N]$, we have $f(x) \geq f(y)$. NSW is clearly in this family, and there are many other standard fairness measures that fall into this class; see Appendix B.2.1. For any SWF in this family, we prove the following regret bound, which remarkably has no dependence on the number of agents $N$ at all.

**Theorem 4.2.** *For any* $f : [0,1]^N \to [0,1]$ *that is concave and Pareto optimal, Algorithm 2 with the log-barrier regularizer* $\psi(p) = -\sum_{i=1}^{K} \log p_i$ *and* $\eta = \sqrt{\frac{K \log T}{T}}$ *guarantees* $\operatorname{Reg}_{\text{adv}} = \mathcal{O}(\sqrt{KT \log T})$.

*Proof Sketch.* Using the concrete form of $\psi$, it is clear that the local norm $\|\nabla f(u_t^\top p_t)\|_{\nabla^{-2} \psi(p_t)}^2$ simplifies to $\sum_{i=1}^{K} p_{t,i}^2 [\nabla f(u_t^\top p_t)]_i^2 \leq \langle p_t, \nabla f(u_t^\top p_t) \rangle^2$, where the inequality is due to $[\nabla f(u_t^\top p_t)]_i \geq 0$ implied by Pareto optimality. Furthermore, by concavity, we have $\langle p_t, \nabla f(u_t^\top p_t) \rangle \leq f(u_t^\top p_t) - f(0) \leq 1$, and thus the local norm at most 1. The rest of the proof is by direct calculation. $\quad\square$

### 4.2.2 FTRL with Tsallis Entropy Regularizer

In fact, when $f = $ NSW, using the special structure of the welfare function, we find yet another regularizer that ensures a small $\mathcal{O}(N)$ local norm, with the benefit of having smaller dependence on $K$ for the penalty term.

**Theorem 4.3.** *For* $f = $ NSW*, Algorithm 2 with the Tsallis entropy regularizer* $\psi(p) = \frac{1 - \sum_{i=1}^{K} p_i^\beta}{1 - \beta}$, $\beta = \frac{2}{N}$, *and the optimal choice of* $\eta$ *guarantees* $\operatorname{Reg}_{\text{adv}} = \widetilde{\mathcal{O}}(K^{\frac{1}{2} - \frac{1}{N}} \sqrt{NT})$.

The proof is more involved and is deferred to Appendix B.2.3. While the regret in Theorem 4.3 suffers polynomial dependence on $N$, it has better dependence on $K$ compared to Theorem 4.2, and is thus more preferable when $K$ is much larger than $N$.

### 4.2.3 Logarithmic Regret for a Special Case

Finally, we discuss a special case with $f = $ NSW where logarithmic regret is possible. This is based on a simple observation that when there is one agent who is indifferent about the learner's choice (that is, the agent's utility is the same for all arms for this round), then $-$NSW is not only convex, but also exp-concave, a stronger curvature property. Therefore, by applying known results, specifically the EWOO algorithm [Hazan et al., 2007], we achieve the following result.

**Theorem 4.4.** *Fix* $f = $ NSW*. Suppose that for each time* $t$*, there is a set of agents* $A_t \subseteq [N]$ *such that* $|A_t| \geq M$ *and* $u_{t,:,n} = c_{t,n} \mathbf{1}$ *with* $c_{t,n} \geq 0$ *for each agent* $n \in A_t$*. Then the EWOO algorithm guarantees* $\operatorname{Reg}_{\text{adv}} = \mathcal{O}\left(\frac{N - M}{M} \cdot K \log T\right)$.

The proof, which verifies the exp-concavity of $-$NSW in this special case, can be found in Appendix B.2.4. We note that the reason that we apply EWOO instead of Online Newton Step, another algorithm discussed in [Hazan et al., 2007] for exp-concave losses, is that the latter requires Lipschizness (which, again, is not satisfied by NSW).

# 5 Conclusion

In this work, motivated by recent research on social welfare maximization for the problem of multi-agent multi-armed bandits, we consider a variant with the arguably more natural version of Nash social welfare as the objective function, and develop multiple algorithms and regret upper/lower bounds in different settings (stochastic versus adversarial and full-information versus bandit feedback). Our results show a sharp separation between our problem and previous settings, including the heavily studied Bandit Convex Optimization problem.

There are many interesting future directions. First, in the stochastic bandit setting, we have only shown the tight dependence on $T$, so what about $K$ and $N$? Second, is there a more general strategy/analysis that works for different social welfare functions (similar to our result in Theorem 4.2)? Taking one step further, similar to the recent research on "omniprediction" [Gopalan et al., 2022], is there one single algorithm that works for a class of social welfare functions simultaneously?

## Acknowledgments and Disclosure of Funding

HL and MZ are supported by NSF Award IIS-1943607.

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

# A Omitted Details in Section 3

In this section, we provide the omitted proofs for the results in Section 3. In Appendix A.1, we provide the proof for Theorem 3.1 and in Appendix A.2, we provide the proof for Theorem 3.2.

## A.1 Proof of Theorem 3.1

To prove Theorem 3.1, we first consider the following two events.

**Event 1.** *For all $t \in \{KN_0 + 1, \ldots, T\}$ and $i \in [K]$,*

$$N_{t,i} \geq N_0 + \frac{1}{2} \sum_{\tau=KN_0}^{t} p_{\tau,i} - 18 \log(KT),$$

*where $N_{t,i}$'s and $p_{t,i}$'s are defined in Algorithm 1.*

**Event 2.** *For all $t \in \{KN_0 + 1, \ldots, T\}$, $i \in [K]$, and $n \in [N]$,*

$$u_{i,n} \leq \widehat{u}_{t,i,n} \leq u_{i,n} + 8\sqrt{\frac{u_{i,n} \log(NKT^2)}{N_{t,i}}} + \frac{15 \log(NKT^2)}{N_{t,i}},$$

*where $N_{t,i}$'s are defined in Algorithm 1.*

As we prove in Lemma A.1 and Lemma A.2, Event 1 and Event 2 hold with probability at least $1 - \frac{1}{T}$. Now we prove Theorem 3.1. For convenience, we restate the theorem as follows.

**Theorem 3.1.** *With $N_0 = 1 + 18 \log KT$, Algorithm 1 guarantees $\mathbb{E}[\mathrm{Reg}_{\mathrm{sto}}] = \widetilde{\mathcal{O}}(K^{\frac{2}{N}} T^{\frac{N-1}{N}} + K)$.*

*Proof.* Let $p^\star = \arg\max_{p \in \Delta_K} \mathrm{NSW}(u^\top p)$. According to a standard regret decomposition for UCB-type algorithms, we know that $\mathrm{Reg}_{\mathrm{sto}}$ can be upper bounded as follows:

$$\mathbb{E}[\mathrm{Reg}_{\mathrm{sto}}]$$

$$= \mathbb{E}\left[\sum_{t=1}^{T} \left(\mathrm{NSW}(u^\top p^\star) - \mathrm{NSW}(u^\top p_t)\right)\right]$$

$$= \mathbb{E}\left[\sum_{t=1}^{T} \left(\mathrm{NSW}(u^\top p^\star) - \mathrm{NSW}(u^\top p_t)\right) \,\middle|\, \text{Event 1 and Event 2 hold}\right] + 2$$

$$\qquad\qquad\qquad\qquad\qquad\qquad\qquad\qquad \text{(according to Lemma A.1 and Lemma A.2)}$$

$$\leq \mathbb{E}\left[\sum_{t=KN_0+1}^{T} \left(\mathrm{NSW}(u^\top p^\star) - \mathrm{NSW}(\widehat{u}_t^\top p^\star)\right) \,\middle|\, \text{Event 1 and Event 2 hold}\right] + KN_0 + 2$$

$$\quad + \mathbb{E}\left[\sum_{t=KN_0+1}^{T} \left(\mathrm{NSW}(\widehat{u}_t^\top p^\star) - \mathrm{NSW}(\widehat{u}_t^\top p_t)\right) \,\middle|\, \text{Event 1 and Event 2 hold}\right]$$

$$\quad + \mathbb{E}\left[\sum_{t=KN_0+1}^{T} \left(\mathrm{NSW}(\widehat{u}_t^\top p_t) - \mathrm{NSW}(u^\top p_t)\right) \,\middle|\, \text{Event 1 and Event 2 hold}\right]$$

$$\leq \mathbb{E}\left[\sum_{t=1}^{T} \left(\mathrm{NSW}(\widehat{u}_t^\top p^\star) - \mathrm{NSW}(\widehat{u}_t^\top p_t)\right)\right] + \mathbb{E}\left[\sum_{t=1}^{T} \left(\mathrm{NSW}(\widehat{u}_t^\top p_t) - \mathrm{NSW}(u^\top p_t)\right)\right] + KN_0 + 2$$

$$\qquad\qquad\qquad\qquad\qquad\qquad\qquad\qquad\qquad\qquad\qquad \text{(based on Event 2)}$$

$$\leq \mathbb{E}\left[\sum_{t=KN_0+1}^{T} \left(\mathrm{NSW}(\widehat{u}_t^\top p_t) - \mathrm{NSW}(u^\top p_t)\right) \,\middle|\, \text{Event 1 and Event 2 hold}\right] + KN_0 + 2.$$

$$\qquad\qquad\qquad\qquad\qquad\qquad\qquad\qquad\qquad\qquad \text{(based on the definition of $p_t$)}$$

In the following, we bound the first term

$$\mathbb{E}\left[\sum_{t=KN_0+1}^{T}\left(\mathrm{NSW}(\widehat{u}_t^\top p_t) - \mathrm{NSW}(u^\top p_t)\right)\ \Big|\ \text{Event 1 and Event 2 hold}\right].$$

As discussed in Section 3.1, we consider two cases. First, consider the set of rounds $\mathcal{T}_\sigma$ such that for all $t \in \mathcal{T}_\sigma$ there exists at least one $n \in [N]$ such that $\langle p_t, u_{:,n}\rangle \le \sigma$ for some $\sigma$ that we will specify later. Denote such $n$ to be $n_t$ (if there are multiple such $n$'s, we pick an arbitrary one). According to Event 2, we know that for all $i \in [K]$,

$$u_{i,n_t} \le \widehat{u}_{t,i,n_t} \le u_{i,n_t} + 8\sqrt{\frac{u_{i,n_t}\log(NKT^2)}{N_{t,i}}} + \frac{15\log(NKT^2)}{N_{t,i}} \le 2u_{i,n_t} + \mathcal{O}\left(\frac{\log(NKT^2)}{N_{t,i}}\right),$$

where the last inequality is because of AM-GM inequality. Therefore, we know that

$$\langle p_t, \widehat{u}_{t,:,n_t}\rangle \le 2\langle p_t, u_{:,n_t}\rangle + \widetilde{\mathcal{O}}\left(\sum_{j=1}^{K}\frac{p_{t,j}}{N_{t,j}}\right) \le 2\sigma + \widetilde{\mathcal{O}}\left(\sum_{j=1}^{K}\frac{p_{t,j}}{N_{t,j}}\right).$$

Now consider $\sum_{t \in \mathcal{T}_\sigma}\left(\mathrm{NSW}(\widehat{u}_t^\top p_t) - \mathrm{NSW}(u^\top p_t)\right)$. Direct calculation shows that

$$\sum_{t\in\mathcal{T}_\sigma}\left(\mathrm{NSW}(\widehat{u}_t^\top p_t) - \mathrm{NSW}(u^\top p_t)\right)$$

$$\le \sum_{t\in\mathcal{T}_\sigma}\mathrm{NSW}(\widehat{u}_t^\top p_t) \qquad\qquad (\text{since } \mathrm{NSW}(u^\top p_t) \ge 0)$$

$$\le \sum_{t\in\mathcal{T}_\sigma}\langle p_t, \widehat{u}_{t,:,n_t}\rangle^{\frac{1}{N}}$$

$$\le 2|\mathcal{T}_\sigma|\cdot\sigma^{\frac{1}{N}} + \sum_{t=1}^{T}\widetilde{\mathcal{O}}\left(\left(\sum_{j=1}^{K}\frac{p_{t,j}}{N_{t,j}}\right)^{\frac{1}{N}}\right) \qquad (\text{since } (a+b)^{\frac{1}{N}} \le a^{\frac{1}{N}} + b^{\frac{1}{N}})$$

$$\le 2|T_\sigma|\cdot\sigma^{\frac{1}{N}} + T^{\frac{N-1}{N}}\widetilde{\mathcal{O}}\left(\left(\sum_{t=1}^{T}\sum_{j=1}^{K}\frac{p_{t,j}}{N_{t,j}}\right)^{\frac{1}{N}}\right) \qquad (\text{Hölder's inequality})$$

$$\le 2T\cdot\sigma^{\frac{1}{N}} + \widetilde{\mathcal{O}}\left(K^{\frac{1}{N}}\cdot T^{\frac{N-1}{N}}\right). \qquad\qquad (\text{using Lemma A.3})$$

Now consider the regret within $t \in \{NK_0+1,\ldots,T\}\backslash\mathcal{T}_\sigma$, in which we have $\langle p_t, u_{:,n}\rangle \ge \sigma$ for all $n \in [N]$. In this case, we bound $\sum_{t\notin\mathcal{T}_\sigma}\left(\mathrm{NSW}(\widehat{u}_t^\top p_t) - \mathrm{NSW}(u^\top p_t)\right)$ as follows:

$$\sum_{t\notin\mathcal{T}_\sigma}\left(\mathrm{NSW}(\widehat{u}_t^\top p_t) - \mathrm{NSW}(u^\top p_t)\right)$$

$$\le \sum_{t\notin\mathcal{T}_\sigma}\sum_{n\in[N]}\left[\langle p_t, \widehat{u}_{t,:,n}\rangle^{\frac{1}{N}} - \langle p_t, u_{:,n}\rangle^{\frac{1}{N}}\right] \qquad (\text{using Lemma C.1 and Event 2})$$

$$= \sum_{t\notin\mathcal{T}_\sigma}\sum_{n\in[N]}\frac{\langle p_t, \widehat{u}_{t,:,n} - u_{:,n}\rangle}{\sum_{k=0}^{N-1}\langle p_t, \widehat{u}_{t,:,n}\rangle^{\frac{k}{N}}\langle p_t, u_{:,n}\rangle^{\frac{N-1-k}{N}}}$$

$$\le \sum_{t\notin\mathcal{T}_\sigma}\sum_{n\in[N]}\frac{\langle p_t, \widehat{u}_{t,:,n} - u_{:,n}\rangle}{N\langle p_t, u_{:,n}\rangle^{\frac{N-1}{N}}} \qquad (\text{since } \widehat{u}_{t,i,n} \ge u_{i,n} \text{ for all } i,t,n \text{ based on Event 2})$$

$$\le \sum_{t\notin\mathcal{T}_\sigma}\sum_{n\in[N]}\frac{\sum_{j=1}^{K}p_{t,j}\left(8\sqrt{\frac{u_{n,j}\log(NKT^2)}{N_{t,j}}} + \frac{8\log(NKT^2)}{N_{t,j}}\right)}{N\langle p_t, u_{:,n}\rangle^{\frac{N-1}{N}}} \qquad (\text{using Event 2})$$

$$\leq \sum_{t\notin\mathcal{T}_\sigma}\sum_{n\in[N]}\left(\frac{\sum_{j=1}^K 8\sqrt{\frac{p_{t,j}\log(NKT^2)}{N_{t,j}}}}{N\langle p_t,u_{:,n}\rangle^{\frac{N-1}{N}-\frac{1}{2}}}+\frac{\sum_{j=1}^K\frac{8p_{t,j}\log(NKT^2)}{N_{t,j}}}{N\langle p_t,u_{:,n}\rangle^{\frac{N-1}{N}}}\right)$$

$$\text{(since }\sqrt{\langle p_t,u_{:,n}\rangle}\geq\sqrt{p_{t,i}u_{i,n}}\text{ for all }i\in[K])$$

$$\leq\widetilde{\mathcal{O}}\left(\frac{1}{N\sigma^{\frac{N-1}{N}-\frac{1}{2}}}\sum_{t\in T}\sum_{n\in[N]}\sum_{j=1}^K\sqrt{\frac{p_{t,j}}{N_{t,j}}}+\frac{1}{N\sigma^{\frac{N-1}{N}}}\sum_{t=1}^T\sum_{n\in[N]}\sum_{j=1}^K\frac{p_{t,j}}{N_{t,j}}\right)$$

$$\leq\widetilde{\mathcal{O}}\left(\sigma^{\frac{1}{2}-\frac{N-1}{N}}K\sqrt{T}+K\cdot\sigma^{-\frac{N-1}{N}}\right),$$

where the last inequality is because Lemma A.3. Combining the regret for both parts, we know that

$$\mathbb{E}[\text{Reg}_{\text{sto}}]\leq\widetilde{\mathcal{O}}\left(K^{\frac{1}{N}}T^{\frac{N-1}{N}}+T\cdot\sigma^{\frac{1}{N}}+\sigma^{\frac{1}{2}-\frac{N-1}{N}}K\sqrt{T}+K\sigma^{-\frac{N-1}{N}}+K\right).$$

Picking the optimal $\sigma$ leads to the expected regret bounded by $\mathbb{E}[\text{Reg}_{\text{sto}}]\leq\widetilde{\mathcal{O}}\left(K^{\frac{2}{N}}T^{\frac{N-1}{N}}+K\right)$.
$\square$

**Lemma A.1.** *Event 1 happens with probability at least* $1-\frac{1}{T}$.

*Proof.* According to Algorithm 1, we know that $N_{(KN_0+1),i}=N_0$ for each $i\in[K]$. Consider the case when $t\geq KN_0+1$. According to Freedman's inequality (Lemma C.3), we have with probability at least $1-\delta$, for a fixed $t\geq KN_0+1$,

$$\sum_{\tau=KN_0+1}^t\mathbb{1}\{i_\tau=i\}\geq\sum_{\tau=KN_0+1}^t p_{\tau,i}-2\sqrt{\sum_{\tau=KN_0}^t p_{\tau,i}\log(1/\delta)}-\log(1/\delta)$$

$$\geq\frac{1}{2}\sum_{\tau=KN_0+1}^t p_{\tau,i}-9\log(1/\delta).$$

Therefore, we know that with probability at least $1-\delta$, for a fixed $t\geq KN_0+1$,

$$N_{t,i}=N_0+\sum_{\tau=KN_0+1}^t\mathbb{1}\{i_\tau=i\}\geq N_0+\frac{1}{2}\sum_{\tau=KN_0+1}^t p_{\tau,i}-9\log(1/\delta).$$

Picking $\delta=\frac{1}{KT^2}$ and taking a union bound over all $i\in[K]$ and $KN_0+1\leq t\leq T$ reach the result.
$\square$

**Lemma A.2.** *Event 2 happens with probability at least* $1-\frac{1}{T}$.

*Proof.* According to Freedman's inequality Lemma C.3, applying a union bound over $t\in[T]$, $i\in[K]$, and $n\in[N]$, we know that with probability at least $1-\delta$, for all $t\in[T]$, $i\in[K]$, and $n\in[N]$,

$$|\bar{u}_{t,i,n}-u_{i,n}|\leq 2\sqrt{\frac{u_{i,n}\log(NKT/\delta)}{N_{t,i}}}+\frac{\log(NKT/\delta)}{N_{t,i}}. \tag{10}$$

Solving the inequality with respect to $u_{i,n}$, we know that

$$\sqrt{u_{i,n}}\leq\sqrt{\frac{\log(NKT/\delta)}{N_{t,i}}}+\sqrt{\bar{u}_{t,i,n}+\frac{2\log(NKT/\delta)}{N_{t,i}}}$$

$$\leq\sqrt{2\bar{u}_{t,i,n}+\frac{6\log(NKT/\delta)}{N_{t,i}}},\qquad\text{(using AM-GM inequality)}$$

$$\bar{u}_{t,i,n}\leq\left(\sqrt{u_{i,n}}+\sqrt{\frac{\log(NKT/\delta)}{N_{t,i}}}\right)^2$$

$$\leq 2u_{i,n} + \frac{2\log(NKT/\delta)}{N_{t,i}}. \qquad \text{(using AM-GM inequality)}$$

Using the above inequality and picking $\delta = \frac{1}{T}$, we can lower bound $\widehat{u}_{t,i,n}$ as follows:

$$\widehat{u}_{t,i,n} = \bar{u}_{t,i,n} + 4\sqrt{\frac{\bar{u}_{t,i,n}\log(NKT^2)}{N_{t,i}}} + \frac{8\log(NKT^2)}{N_{t,i}}$$

$$\geq \bar{u}_{t,i,n} + 4\sqrt{\frac{\log(NKT^2)}{N_{t,i}}\left(\frac{u_{i,n}}{2} - \frac{3\log(NKT^2)}{N_{t,i}}\right)} + \frac{8\log(NKT^2)}{N_{t,i}}$$

$$\geq \bar{u}_{t,i,n} + 4\sqrt{\frac{u_{i,n}\log(NKT^2)}{2N_{t,i}}} + \frac{(8-4\sqrt{3})\log(NKT^2)}{N_{t,i}}$$

$$\text{(using } \sqrt{a-b} \geq \sqrt{a} - \sqrt{b} \text{ for } a, b \geq 0)$$

$$\geq \bar{u}_{t,i,n} + 2\sqrt{\frac{u_{i,n}\log(NKT^2)}{N_{t,i}}} + \frac{\log(NKT^2)}{N_{t,i}}$$

$$\geq u_{i,n},$$

where the last inequality uses Eq. (10). To upper bound $\widehat{u}_{t,i,n}$, we have

$$\widehat{u}_{t,i,n} = \bar{u}_{t,i,n} + 4\sqrt{\frac{\bar{u}_{t,i,n}\log(NKT^2)}{N_{t,i}}} + \frac{8\log(NKT^2)}{N_{t,i}}$$

$$\leq u_{i,n} + 2\sqrt{\frac{u_{i,n}\log(NKT^2)}{N_{t,i}}} + \frac{\log(NKT^2)}{N_{t,i}} \qquad \text{(using Eq. (10))}$$

$$+ 4\sqrt{\frac{\log(NKT^2)}{N_{t,i}}\left(2u_{i,n} + \frac{2\log(NKT^2)}{N_{t,i}}\right)} + \frac{8\log(NKT^2)}{N_{t,i}}$$

$$\leq u_{i,n} + 8\sqrt{\frac{u_{i,n}\log(NKT^2)}{N_{t,i}}} + \frac{15\log(NKT^2)}{N_{t,i}}. \qquad \text{(using AM-GM inequality)}$$

Combining the lower and the upper bound finishes the proof. $\qquad\square$

**Lemma A.3.** *Under Event 1, Algorithm 1 guarantees that*

$$\sum_{\tau=KN_0+1}^{t} \frac{p_{\tau,i}}{N_{\tau,i}} \leq \mathcal{O}\left(\log T\right),$$

$$\sum_{\tau=KN_0+1}^{t} \sqrt{\frac{p_{\tau,i}}{N_{\tau,i}}} \leq \mathcal{O}\left(\sqrt{T\log T}\right),$$

*for all $i \in [K]$ and $KN_0 + 1 \leq t \leq T$.*

*Proof.* Since Event 1 holds, we know that $N_{t,i} \geq \frac{1}{2}\sum_{\tau=KN_0+1}^{t} p_{\tau,i} + 1$ holds for all $i \in [K]$ and $\tau \geq KN_0 + 1$ based on the choice of $N_0 = 18\log KT + 1$. Therefore, we know that for each $t \geq KN_0 + 1$,

$$\sum_{\tau=KN_0+1}^{t} \frac{p_{\tau,i}}{N_{\tau,i}} \leq \sum_{\tau=KN_0+1}^{t} \frac{2p_{\tau,i}}{\sum_{\tau'=KN_0+1}^{\tau} p_{\tau',i} + 2}$$

$$\leq 2\int_{0}^{\sum_{\tau=KN_0+1}^{t} p_{\tau,i}} \frac{1}{x+2}dx \leq 2\log(T+2).$$

As for the term $\sum_{\tau=KN_0+1}^{t} \sqrt{\frac{p_{\tau,i}}{N_{\tau,i}}}$, we have

$$\sum_{\tau=KN_0+1}^{t} \sqrt{\frac{p_{\tau,i}}{N_{\tau,i}}} \leq \sqrt{(t-KN_0)\sum_{\tau=KN_0+1}^{t} \frac{p_{\tau,i}}{N_{\tau,i}}} \qquad \text{(Cauchy-Schwarz inequality)}$$

$$\le \mathcal{O}(\sqrt{T \log T}).$$

$\square$

## A.2 Omitted Details in Section 3.2

**Theorem 3.2.** *In the bandit feedback setting, for any algorithm, there exists a stochastic environment in which the expected regret (with respect to* NSW*) of this algorithm is* $\Omega\left(\frac{(\log K)^3}{N^3} \cdot K^{\frac{1}{N}} T^{\frac{N-1}{N}}\right)$ *for* $N \ge \log K$ *and sufficiently large* $T$.

*Proof.* Consider the environment $\mathcal{E}$ that picks $u$ uniformly from $\{u^{(1)}, \dots, u^{(K)}\}$, where $u^{(i)}_{:,1} = \varepsilon \cdot \mathbf{e}_i \in \mathbb{R}^K$ and $u^{(i)}_{:,j} = \mathbf{1}$ for all $j \in \{2, 3, \dots, N\}$. Here, $\varepsilon \in (0, \frac{1}{9}]$ is some constant to be specified later. Denote $u^{(0)}$ to be the environment where $u_{:,n} = \mathbf{0}$ for all $n \in [N]$. At each round $t$, $u_{t,i,n}$ is an i.i.d. Bernoulli random variable with mean $u_{i,n}$. For notational convenience, we use $\mathbb{E}_i[\cdot]$ when we take expectation over the environment $u^{(i)}$ for $i \in \{0\} \cup [K]$. Let $n_i$ be the number of rounds that action $i$ is selected over the total horizon $T$ for all $i \in [K]$. Therefore, the expected regret with respect to environment $\mathcal{E}$ (a uniform distribution over $u^{(i)}$, $i \in [K]$) is lower bounded as follows:

$$\mathbb{E}_{\mathcal{E}}[\text{Reg}] = \frac{1}{K} \sum_{i=1}^{K} \mathbb{E}_i \left[ \varepsilon^{\frac{1}{N}} \sum_{t=1}^{T} (1 - p_{t,i}^{\frac{1}{N}}) \right]$$

$$\ge T\varepsilon^{\frac{1}{N}} - \frac{\varepsilon^{\frac{1}{N}} T^{\frac{N-1}{N}}}{K} \sum_{i=1}^{K} \mathbb{E}_i \left[ \left( \sum_{t=1}^{T} p_{t,i} \right)^{\frac{1}{N}} \right] \qquad \text{(Hölder's inequality)}$$

$$\ge T\varepsilon^{\frac{1}{N}} - \frac{\varepsilon^{\frac{1}{N}} T^{\frac{N-1}{N}}}{K} \sum_{i=1}^{K} \left( \mathbb{E}_i \left[ \sum_{t=1}^{T} p_{t,i} \right] \right)^{\frac{1}{N}} \qquad \text{(Jensen's inequality)}$$

$$\ge T\varepsilon^{\frac{1}{N}} - K^{-\frac{1}{N}} \varepsilon^{\frac{1}{N}} T^{\frac{N-1}{N}} \left( \sum_{i=1}^{K} \mathbb{E}_i \left[ \sum_{t=1}^{T} p_{t,i} \right] \right)^{\frac{1}{N}}, \qquad (11)$$

where the last inequality is again due to Hölder's inequality. Let $\text{Ber}(\alpha)$ be the Bernoulli distribution with mean $\alpha$. Combining Exercise 14.4 of [Lattimore and Szepesvári, 2020], Pinsker's inequality, and Lemma 15.1 of [Lattimore and Szepesvári, 2020], we have

$$\mathbb{E}_i \left[ \sum_{t=1}^{T} p_{t,i} \right] = \mathbb{E}_i[n_i] \le \mathbb{E}_0[n_i] + T \sqrt{\frac{1}{2} \mathbb{E}_0[n_i] \text{KL}(\text{Ber}(0)|\text{Ber}(\varepsilon))}$$

$$\le \mathbb{E}_0[n_i] + T \sqrt{\frac{1}{2} \mathbb{E}_0[n_i] \log\left(1 + \frac{\varepsilon}{1 - \varepsilon}\right)}$$

$$\le \mathbb{E}_0[n_i] + T \sqrt{\mathbb{E}_0[n_i] \frac{\varepsilon}{2(1 - \varepsilon)}} \qquad \text{(using } \log(1 + x) \le x\text{)}$$

$$\le \mathbb{E}_0[n_i] + \frac{3T}{4} \sqrt{\mathbb{E}_0[n_i] \varepsilon}$$

$$= \mathbb{E}_0 \left[ \sum_{t=1}^{T} p_{t,i} \right] + \frac{3T}{4} \sqrt{\mathbb{E}_0 \left[ \sum_{t=1}^{T} p_{t,i} \right] \varepsilon}. \qquad (12)$$

where the last inequality is because $\varepsilon \le \frac{1}{9}$.

Taking summation over all $i \in [K]$, we obtain that

$$\sum_{i \in [K]} \mathbb{E}_i \left[ \sum_{t=1}^{T} p_{t,i} \right]$$

$$\leq \sum_{i \in [K]} \mathbb{E}_0 \left[ \sum_{t=1}^{T} p_{t,i} \right] + \frac{3}{4} T \sum_{i=1}^{K} \sqrt{\mathbb{E}_0 \left[ \sum_{t=1}^{T} p_{t,i} \right]} \varepsilon$$

$$\leq T + \frac{3T}{4} \sqrt{K \varepsilon \mathbb{E}_0 \left[ \sum_{i=1}^{K} \sum_{t=1}^{T} p_{t,i} \right]}$$

$$= T + \frac{3T}{4} \sqrt{KT\varepsilon} \tag{13}$$

where the second inequality is due to Cauchy-Schwarz inequality.

Applying Eq. (13) to Eq. (11), we obtain that

$$\mathbb{E}_{\mathcal{E}}[\text{Reg}]$$

$$\geq T\varepsilon^{\frac{1}{N}} - K^{-\frac{1}{N}} \varepsilon^{\frac{1}{N}} T^{\frac{N-1}{N}} \left( T + \frac{3T}{4} \sqrt{KT\varepsilon} \right)^{\frac{1}{N}}$$

$$\geq \left( 1 - K^{-\frac{1}{N}} \right) T\varepsilon^{\frac{1}{N}} - K^{-\frac{1}{2N}} \varepsilon^{\frac{3}{2N}} T^{\frac{2N+1}{2N}} \qquad \text{(using } (a+b)^{\frac{1}{N}} \leq a^{\frac{1}{N}} + b^{\frac{1}{N}} \text{)}$$

$$\geq \frac{\log K}{2N} T\varepsilon^{\frac{1}{N}} - K^{-\frac{1}{2N}} \varepsilon^{\frac{3}{2N}} T^{\frac{2N+1}{2N}},$$

where the third inequality is according to Lemma C.2 with $x = \frac{1}{N}$ and $\alpha = \frac{1}{K}$, meaning that $N \left( 1 - \frac{1}{K}^{\frac{1}{N}} \right) \geq \frac{\log K}{2}$.

Picking $\varepsilon = \frac{(\log K)^{2N} \cdot K}{(4N)^{2N} T}$, we know that

$$K^{-\frac{1}{2N}} \varepsilon^{\frac{3}{2N}} T^{\frac{2N+1}{2N}} = \frac{\varepsilon^{\frac{1}{N}} T \log K}{4N} = \Omega \left( \frac{(\log K)^3}{N^3} \cdot K^{\frac{1}{N}} T^{\frac{N-1}{N}} \right),$$

Combining the above all together, we know that $\mathbb{E}_{\mathcal{E}}[\text{Reg}] \geq \Omega \left( \frac{(\log K)^3}{N^3} \cdot K^{\frac{1}{N}} T^{\frac{N-1}{N}} \right)$. Therefore, there exists one environment among $u^{(i)}, i \in [K]$ such that $\mathbb{E}_i[\text{Reg}] \geq \Omega \left( \frac{(\log K)^3}{N^3} \cdot K^{\frac{1}{N}} T^{\frac{N-1}{N}} \right)$, which finishes the proof. $\square$

# B  Omitted Details in Section 4

## B.1  Omitted Details in Section 4.1

In this section, we prove that that in the adversarial environment, it is also impossible to achieve sublinear regret when $f = \text{NSW}_{\text{prod}}$. The hard instance construction shares a similar spirit to the one for $f = \text{NSW}$ shown in Theorem 4.1.

**Theorem B.1.** *In the bandit feedback setting, for any algorithm, there exists an adversarial environment such that* $\mathbb{E}[\text{Reg}_{\text{adv}}] = \Omega(T)$ *for* $f = \text{NSW}_{\text{prod}}$.

*Proof.* We consider the learning environment with two agents and two arms. The agents utilities are binary, meaning that $u \in \{0,1\}^{2 \times 2}$. We construct two distributions $\mathcal{E}$ and $\mathcal{E}'$ with support $\{0,1\}^{2 \times 2}$. To define environment $\mathcal{E}$, we use $q_{wxyz}$ for any $w, x, y, z \in \{0,1\}$ to denote the probability that $u_{1,:} = (w, x)$ and $u_{2,:} = (y, z)$ when $u \sim \mathcal{E}$. For simplicity of notation, the binary number $wxyz$ will be written in decimal form (i.e. $q_8 = \text{Pr}_{u \sim \mathcal{E}}[u_{1,:} = (1,0), u_{2,:} = (0,0)]$). For environment $\mathcal{E}$, we assign the probabilities

$$q_i = \frac{1}{16} \quad \text{for } i \in \{0, \ldots, 15\} \setminus \{0, 2, 4, 6\} \qquad (q_0, q_2, q_4, q_6) = \left( \frac{1}{8}, 0, 0, \frac{1}{8} \right)$$

Similarly, for environment $\mathcal{E}'$, we use $q'_{wxyz}$ for any $w, x, z, y \in \{0,1\}$ to denote the probability that $u'_{1,:} = (w, x)$ and $u'_{2,:} = (y, z)$ when $u' \sim \mathcal{E}'$. Again, we will write the binary number $wxyz$ in decimal form for ease of notation. To environment $\mathcal{E}'$ we assign probabilities

$$q'_i = \frac{1}{16} \quad \text{for } i \in \{0, \ldots, 15\} \setminus \{1, 3, 5, 7\} \qquad (q'_1, q'_3, q'_5, q'_7) = \left( 0, \frac{1}{8}, \frac{1}{8}, 0 \right)$$

Next, we argue that the learner's observations are equivalent in distribution in $\mathcal{E}$ and $\mathcal{E}'$, since the marginal distribution of every possible observation of each action is the same. Specifically,

- When action 1 is played, the learner's observation $u_{1,:}$ in both $\mathcal{E}$ and $\mathcal{E}'$ are given by the following marginal distribution.
  - The probability of observation $(0,0)$ is $q_0 + q_1 + q_2 + q_3 = q'_0 + q'_1 + q'_2 + q'_3 = \frac{1}{4}$
  - The probability of observation $(0,1)$ is $q_4 + q_5 + q_6 + q_7 = q'_4 + q'_5 + q'_6 + q'_7 = \frac{1}{4}$
  - The probability of observation $(1,0)$ is $q_8 + q_9 + q_{10} + q_{11} = q'_8 + q'_9 + q'_{10} + q'_{11} = \frac{1}{4}$
  - The probability of observation $(1,1)$ is $q_{12} + q_{13} + q_{14} + q_{15} = q'_{12} + q'_{13} + q'_{14} + q'_{15} = \frac{1}{4}$

- When action 2 is played, the learner's observation $u_{2,:}$ in both $\mathcal{E}$ and $\mathcal{E}'$ are given by the following marginal distribution.
  - The probability of observation $(0,0)$ is $q_0 + q_4 + q_8 + q_{12} = q'_0 + q'_4 + q'_8 + q'_{12} = \frac{1}{4}$
  - The probability of observation $(0,1)$ is $q_1 + q_5 + q_9 + q_{13} = q'_1 + q'_5 + q'_9 + q'_{13} = \frac{1}{4}$
  - The probability of observation $(1,0)$ is $q_2 + q_6 + q_{10} + q_{14} = q'_2 + q'_6 + q'_{10} + q'_{14} = \frac{1}{4}$
  - The probability of observation $(1,1)$ is $q_3 + q_7 + q_{11} + q_{15} = q'_3 + q'_7 + q'_{11} + q'_{15} = \frac{1}{4}$

Direct calculation shows that

$$\mathbb{E}_{u \sim \mathcal{E}} \left[ \text{NSW}_{\text{prod}}(u^\top p) \right]$$
$$= (q_5 - q_6 - q_9 + q_{10})p_1^2 + (q_6 - q_7 - 2q_5 + q_9 + q_{11} - q_{13} + q_{14})p_1 + (q_5 + q_7 + q_{13} + q_{15})$$
$$= -\frac{1}{16}p_1^2 + \frac{1}{16}p_1 + \frac{1}{4},$$
$$\mathbb{E}_{u \sim \mathcal{E}'} \left[ \text{NSW}_{\text{prod}}(u^\top p) \right]$$
$$= (q'_5 - q'_6 - q'_9 + q'_{10})p_1^2 + (q'_6 - q'_7 - 2q'_5 + q'_9 + q'_{11} - q'_{13} + q'_{14})p_1 + (q'_5 + q'_7 + q'_{13} + q'_{15})$$
$$= \frac{1}{16}p_1^2 - \frac{1}{16}p_1 + \frac{1}{4}.$$

Therefore, we compute the learner's best strategy for environments $\mathcal{E}$ and $\mathcal{E}'$ by direct calculation:

$$p_\star = \underset{p \in \Delta_2}{\arg\max} \, \mathbb{E}_{u \sim \mathcal{E}} \left[ \text{NSW}_{\text{prod}}(u^\top p) \right] = \underset{p \in \Delta_2}{\arg\max} \left[ \frac{1}{4} + \frac{1}{16}p_1 - \frac{1}{16}p_1^2 \right] = \left( \frac{1}{2}, \frac{1}{2} \right)$$

$$p'_\star = \underset{p \in \Delta_2}{\arg\max} \, \mathbb{E}_{u \sim \mathcal{E}'} \left[ \text{NSW}_{\text{prod}}(u^\top p) \right] = \underset{p \in \Delta_2}{\arg\max} \left[ \frac{1}{4} - \frac{1}{16}p_1 + \frac{1}{16}p_1^2 \right] = \{(0,1), (1,0)\}$$

Next, consider a distribution $p \in \Delta_2$ such that $p_1 \in \left[0, \frac{1}{4}\right] \cup \left[\frac{3}{4}, 1\right]$. For such $p$, direct calculation shows that $\mathbb{E}_{u \sim \mathcal{E}} \left[ \text{NSW}_{\text{prod}}(u^\top p_\star) - \text{NSW}_{\text{prod}}(u^\top p) \right] \geq \Delta$, where $\Delta = \frac{1}{256}$. On the other hand, for a strategy $p \in \Delta_2$ with $p_1 \in \left(\frac{1}{4}, \frac{3}{4}\right)$, we have $\mathbb{E}_{u \sim \mathcal{E}'} \left[ \text{NSW}_{\text{prod}}(u^\top p'_\star) - \text{NSW}_{\text{prod}}(u^\top p) \right] > \Delta$ as well. Given any algorithm, let $\alpha_{\mathcal{E}}$ be the probability that the number of rounds $p_{t,1} \in \left[0, \frac{1}{4}\right] \cup \left[\frac{3}{4}, 1\right]$ is larger than $\frac{T}{2}$ under environment $\mathcal{E}$. Let $\bar{\alpha}_{\mathcal{E}'}$ be the probability of the complement of this event under environment $\mathcal{E}'$. By definition

$$\mathbb{E}_{\mathcal{E}}[\text{Reg}_{\text{adv}}] \geq \mathbb{E}_{\mathcal{E}} \left[ \sum_{t=1}^{T} \text{NSW}_{\text{prod}}(u_t^\top p_\star) - \sum_{t=1}^{T} \text{NSW}_{\text{prod}}(u_t^\top p_t) \right] \geq \frac{\alpha_{\mathcal{E}} T \Delta}{2}$$

$$\mathbb{E}_{\mathcal{E}'}[\text{Reg}_{\text{adv}}] \geq \mathbb{E}_{\mathcal{E}'} \left[ \sum_{t=1}^{T} \text{NSW}_{\text{prod}}(u_t^\top p'_\star) - \sum_{t=1}^{T} \text{NSW}_{\text{prod}}(u_t^\top p_t) \right] \geq \frac{\bar{\alpha}_{\mathcal{E}'} T \Delta}{2}$$

Since the feedback for the algorithm is the same in distribution, we have $\alpha_{\mathcal{E}} + \bar{\alpha}_{\mathcal{E}'} = 1$. Thus, we have

$$\max\{\mathbb{E}_{\mathcal{E}}[\text{Reg}_{\text{adv}}], \mathbb{E}_{\mathcal{E}'}[\text{Reg}_{\text{adv}}]\} \geq \frac{\mathbb{E}_{\mathcal{E}}[\text{Reg}_{\text{adv}}] + \mathbb{E}_{\mathcal{E}'}[\text{Reg}_{adv}]}{2} \geq \frac{(\alpha_{\mathcal{E}} + \bar{\alpha}_{\mathcal{E}'})T\Delta}{4} = \Omega(T).$$

$\square$

## B.2 Omitted Details in Section 4.2

### B.2.1 Concave and Pareto Optimal SWFs

Formally, for a function $f : [0,1]^N \mapsto [0,1]$, concavity and Pareto Optimality are defined as:

- Concavity: $f(\alpha x + (1-\alpha)y) \leq \alpha f(x) + (1-\alpha)f(y)$ for any $\alpha \in [0,1]$ and $x, y \in [0,1]^N$.
- Pareto optimality: for any $x, y \in [0,1]^N$, $x_n \geq y_n$ for all $n \in [N]$ implies $f(x) \geq f(y)$.

Pareto optimality is a "fundamental property" in social choice theory because it ensures that a SWF prefers alternatives that strictly more efficient: everyone is no worse off [Kaneko and Nakamura, 1979]. Concavity appears less in social choice literature. However, it promotes equity by modeling diminishing levels of desirability with the increase of a single agent's utility.

In the following, we provide examples of SWFs that satisfy concavity and Pareto optimality. Each of the following SWFs are parameterized by fixed weights $w \in \Delta_K$.

- Utilitarian SWF: $f(u) = \langle w, u \rangle$;
- Generalized Gini Index (GGI): $f(u) = \min_{\pi \in \mathbb{S}_N} \langle w_\pi, u \rangle$, where $\mathbb{S}_N$ is the set of permutations over $N$ items and $w_\pi$ is weights $w$ permuted according to $\pi \in \mathbb{S}_N$;
- Weighted NSW: $f(u) = \prod_{n \in N} u_n^{w_n}$.

The last notable fact about the class of concave and Pareto Optimal SWFs is that it is closed under convex combinations. Specifically, for two concave and Pareto Optimal functions $f, g : [0,1]^N \to [0,1]$, the function $h(\cdot) = \lambda f(\cdot) + (1-\lambda)g(\cdot)$ for any $\lambda \in [0,1]$ is concave and Pareto Optimal. [Chen and Hooker, 2023] discusses how such convex combinations can be used to combine a SWF prioritizing efficiency and another prioritizing equity to derive a different SWF that prioritizes a balance between efficiency and equity.

### B.2.2 Omitted Details in Section 4.2.1

In this section, we prove Theorem 4.2, which shows that $\mathcal{O}(\sqrt{KT \log T})$ regret is achievable for all concave and Pareto optimal SWFs.

**Theorem 4.2.** *For any $f : [0,1]^N \to [0,1]$ that is concave and Pareto optimal, Algorithm 2 with the log-barrier regularizer $\psi(p) = -\sum_{i=1}^K \log p_i$ and $\eta = \sqrt{\frac{K \log T}{T}}$ guarantees* $\mathrm{Reg}_{\mathrm{adv}} = \mathcal{O}(\sqrt{KT \log T})$.

*Proof.* Using the concavity of $f$, we can upper bound $\mathrm{Reg}_{\mathrm{adv}}$ as follows:

$$
\mathrm{Reg}_{\mathrm{adv}} = \max_{p \in \Delta_K} \sum_{t=1}^T f(u_t^\top p) - \sum_{t=1}^T f(u_t^\top p_t)
$$

$$
\leq \underbrace{\max_{p \in \Delta_{K, \frac{1}{KT}}} \sum_{t=1}^T \langle -\nabla f(u_t^\top p_t), p_t - p \rangle}_{\text{Term (1)}} + \underbrace{\max_{p \in \Delta_K} \sum_{t=1}^T f(u_t^\top p) - \max_{p \in \Delta_{K, \frac{1}{KT}}} \sum_{t=1}^T f(u_t^\top p)}_{\text{Term (2)}},
$$

where $\Delta_{K, \frac{1}{KT}} = \{p \in \Delta_K \mid p_i \geq \frac{1}{KT}, \forall i \in [K]\}$.

To bound Term (1), according to a standard analysis of FTRL/OMD with log-barrier regularizer (e.g. Lemma 12 of [Agarwal et al., 2017]), we know that:

$$
\text{Term (1)} \leq \max_{p \in \Delta_{K, \frac{1}{KT}}} \frac{D_\psi(p, p_1)}{\eta} + \eta \sum_{t=1}^T \sum_{i=1}^K p_{t,i}^2 \cdot \left[ \nabla f(u_t^\top p_t) \right]_i^2, \tag{14}
$$

where $D_\psi(p, q) \triangleq \psi(p) - \psi(q) - \langle \nabla \psi(q), p - q \rangle$ is the Bregman divergence between $p$ and $q$ with respect to $\psi$. To further bound the right-hand side, note that $p_1 = \frac{1}{K} \cdot \mathbf{1}$. Direct calculation shows that for any $p \in \Delta_{K, \frac{1}{KT}}$,

$$
D_\psi(p, p_1) = \sum_{i=1}^K \left( \frac{p_i}{p_{1,i}} - 1 - \log \left( \frac{p_i}{p_{1,i}} \right) \right)
$$

$$= \sum_{i=1}^{K} \log \left( \frac{1}{K p_i} \right)$$

$$\leq K \log \left( \frac{1}{K \cdot \frac{1}{KT}} \right) \qquad \text{(since } p_i \geq \frac{1}{KT} \text{ for all } i \in [K]\text{)}$$

$$\leq K \log T. \tag{15}$$

Using the Pareto optimality property of $f$ and the positivity of the utility matrix $u_t$, we know that $[\nabla f(u_t^\top p)]_i \geq 0$, meaning that $\sum_{i=1}^{K} p_{t,i}^2 \left[ \nabla f(u_t^\top p_t) \right]_i^2 \leq \left\langle p_t, \nabla f(u_t^\top p_t) \right\rangle^2$.

Moreover, using the concavity property of $f$, we know that $\left\langle p_t, \nabla f(u_t^\top p_t) \right\rangle \leq f(u_t^\top p_t) - f(u_t^\top \mathbf{0}) = f(u_t^\top p_t) \leq 1$. Combining the above two inequalities means that

$$\sum_{t=1}^{T} \sum_{i=1}^{K} p_{t,i}^2 \left[ \nabla f(u_t^\top p_t) \right]_i^2 \leq T. \tag{16}$$

Combining Eq. (15) and Eq. (16), we can upper bound `Term` (1) as follows:

$$\texttt{Term } (1) \leq \frac{K \log T}{\eta} + \eta T. \tag{17}$$

Denote the optimal distribution $p^\star = \operatorname{argmax}_{p \in \Delta_K} \sum_{t=1}^{T} f(u_t^\top p)$. Recall that $p_1 = \frac{1}{K} \cdot \mathbf{1}$. Now we upper bound `Term` (2) as follows:

$$\texttt{Term } (2) = \sum_{t=1}^{T} f(u_t^\top p^\star) - \max_{p \in \Delta_{K, \frac{1}{KT}}} \sum_{t=1}^{T} f(u_t^\top p)$$

$$\leq \sum_{t=1}^{T} f(u_t^\top p^\star) - \sum_{t=1}^{T} f\left( u_t^\top \left( \left( 1 - \frac{1}{T} \right) p^\star + \frac{1}{T} \cdot p_1 \right) \right)$$

$$\qquad \qquad ((1 - \tfrac{1}{T}) p^\star + \tfrac{1}{T} p_1 \in \Delta_{K, \frac{1}{KT}})$$

$$\leq \sum_{t=1}^{T} f(u_t^\top p^\star) - \sum_{t=1}^{T} \left[ \left( 1 - \frac{1}{T} \right) \cdot f(u_t^\top p^\star) + \frac{1}{T} \cdot f(u_t^\top p_1) \right] \qquad \text{(Concavity)}$$

$$\leq \frac{1}{T} \cdot \sum_{t=1}^{T} f(u_t^\top p^\star) \qquad \text{(since } f(u_t^\top p_1) \geq 0\text{)}$$

$$\leq 1. \tag{18}$$

Combining Eq. (17) and Eq. (18), and choosing $\eta = \sqrt{\frac{K \log T}{T}}$ finishes the proof. $\qquad \square$

### B.2.3 Omitted Details in Section 4.2.2

In this section, we present the omitted proof for Theorem 4.3, which shows a better dependency on $K$ compared with Theorem 4.2.

**Theorem 4.3.** *For $f = $ NSW, Algorithm 2 with the Tsallis entropy regularizer $\psi(p) = \frac{1 - \sum_{i=1}^{K} p_i^\beta}{1 - \beta}$, $\beta = \frac{2}{N}$, and the optimal choice of $\eta$ guarantees $\mathrm{Reg}_{\mathrm{adv}} = \widetilde{\mathcal{O}}(K^{\frac{1}{2} - \frac{1}{N}} \sqrt{NT})$.*

*Proof.* Consider the case when $N \geq 3$. Direct calculation shows that

$$\left[ \nabla f(u^\top p) \right]_i \leq \frac{1}{N} \sum_{n=1}^{N} \frac{u_{i,n}}{\langle p, u_{:,n} \rangle^{1 - \frac{1}{N}}}. \tag{19}$$

Using the concavity of $f$ and a standard analysis of FTRL with Tsallis entropy (e.g., [Luo, 2017, Theorem 1]), we know that

$$\mathrm{Reg}_{\mathrm{adv}} = \sum_{t=1}^{T} \left( f(u_t^\top p^\star) - f(u_t^\top p_t) \right)$$

$$\leq \langle \nabla f(u_t^\top p_t), p^\star - p_t \rangle \qquad\qquad\qquad\text{(concavity of } f)$$

$$\leq \frac{K^{1-\beta}-1}{\eta(1-\beta)} + \frac{\eta}{\beta} \sum_{t=1}^{T} \sum_{i=1}^{K} p_{t,i}^{2-\beta} \left[ \nabla f(u_t^\top p_t) \right]_i^2$$

$$= \frac{K^{1-\beta}-1}{\eta(1-\beta)} + \frac{\eta}{N^2\beta} \sum_{t=1}^{T} \sum_{i=1}^{K} \left( \sum_{n=1}^{N} \frac{p_{t,i}^{1-\frac{\beta}{2}} u_{t,i,n}}{(\sum_{j=1}^{K} u_{t,j,n} \cdot p_{t,j})^{1-\frac{1}{N}}} \right)^2 \qquad \text{(using Eq. (19))}$$

$$\leq \frac{K^{1-\beta}-1}{\eta(1-\beta)} + \frac{\eta}{N\beta} \sum_{t=1}^{T} \sum_{i=1}^{K} \sum_{n=1}^{N} \frac{p_{t,i}^{2-\beta} u_{t,i,n}^2}{(\sum_{j=1}^{K} u_{t,j,n} \cdot p_{t,j})^{2-\frac{2}{N}}} \qquad \text{(Cauchy-Schwarz inequality)}$$

$$\leq \frac{K^{1-\beta}-1}{\eta(1-\beta)} + \frac{\eta}{N\beta} \sum_{t=1}^{T} \sum_{n=1}^{N} \frac{\sum_{i=1}^{K} p_{t,i}^{2-\beta} u_{t,i,n}^2}{\sum_{j=1}^{K} p_{t,j}^{2-\frac{2}{N}} u_{t,j,n}^{2-\frac{2}{N}}} \qquad \text{(since } (\sum_i x_i)^\alpha \geq \sum_i x_i^\alpha \text{ for } \alpha \geq 1)$$

$$\leq \frac{K^{1-\beta}}{\eta(1-\beta)} + \frac{\eta}{N\beta} \sum_{t=1}^{T} \sum_{n=1}^{N} \frac{\sum_{i=1}^{K} p_{t,i}^{2-\beta} u_{t,i,n}^{2-\frac{2}{N}}}{\sum_{j=1}^{K} p_{t,j}^{2-\frac{2}{N}} u_{t,j,n}^{2-\frac{2}{N}}}. \qquad \text{(since } u_{t,i,n} \in [0,1])$$

Picking $\beta = \frac{2}{N}$, the first term can be upper bounded by $\frac{3K^{1-\frac{2}{N}}}{\eta}$ and the second term can be upper bounded by $\frac{\eta T}{\beta} = \frac{\eta NT}{2}$. Further picking the optimal choice of $\eta$ finishes the proof.

When $N = 2$ and $\beta = \frac{2}{N} = 1$, the regularizer $\psi(p) = \frac{1-\sum_{i=1}^{K} p_i^\beta}{1-\beta}$ becomes the negative Shannon entropy $\psi(p) = \sum_{i=1}^{K} p_i \log p_i$. Using the concavity of $f$ and following a standard analysis of FTRL with Shannon entropy regularizer (e.g., [Hazan et al., 2016, Theorem 5.2]), we obtain that

$$\text{Reg}_{\text{adv}} = \sum_{t=1}^{T} \left( f(u_t^\top p^\star) - f_t(u_t^\top p_t) \right)$$

$$\leq \sum_{t=1}^{T} \langle \nabla f(u_t^\top p_t), p^\star - p_t \rangle \qquad\qquad \text{(using the concavity of } f)$$

$$\leq \frac{\psi(p^\star) - \psi(p_1)}{\eta} + 2\eta \sum_{t=1}^{T} \sum_{i=1}^{K} p_{t,i} \left[ \nabla f(u_t^\top p_t) \right]_i^2 \qquad \text{(by [Hazan et al., 2016, Theorem 5.2])}$$

$$= \frac{\log K}{\eta} + 2\eta \sum_{t=1}^{T} \sum_{i=1}^{K} p_{t,i} \left( \frac{u_{t,i,1}}{2\sqrt{\langle p_t, u_{t,:,1} \rangle}} + \frac{u_{t,i,2}}{2\sqrt{\langle p_t, u_{t,:,2} \rangle}} \right)^2$$

$$\leq \frac{\log K}{\eta} + \eta \sum_{t=1}^{T} \left( \frac{\sum_{i=1}^{K} p_{t,i} u_{t,i,1}^2}{\sum_{i=1}^{K} p_{t,i} u_{t,i,1}} + \frac{\sum_{i=1}^{K} p_{t,i} u_{t,i,2}^2}{\sum_{i=1}^{K} p_{t,i} u_{t,i,2}} \right) \qquad \text{(AM-GM inequality)}$$

$$\leq \frac{\log K}{\eta} + 2\eta T. \qquad\qquad \text{(since } u_{t,i,n} \in [0,1] \text{ for all } t, i, n)$$

Picking $\eta = \sqrt{\frac{\log K}{T}}$ shows that $\text{Reg}_{\text{adv}} = \mathcal{O}\left(\sqrt{T \log K}\right) = \widetilde{\mathcal{O}}(\sqrt{T})$ for $N = 2$. $\qquad\square$

### B.2.4  Omitted Details in Section 4.2.3

In this section, we show the proof for Theorem 4.4, which shows that logarithmic regret is achievable when there is at least one agent who is indifferent about the learner's choice.

**Theorem 4.4.** *Fix $f = $ NSW. Suppose that for each time $t$, there is a set of agents $A_t \subseteq [N]$ such that $|A_t| \geq M$ and $u_{t,:,n} = c_{t,n}\mathbf{1}$ with $c_{t,n} \geq 0$ for each agent $n \in A_t$. Then the EWOO algorithm guarantees $\text{Reg}_{\text{adv}} = \mathcal{O}\left(\frac{N-M}{M} \cdot K \log T\right)$.*

*Proof.* To show that EWOO algorithm achieves logarithmic regret, we need to show that $f_t(p) \triangleq -\text{NSW}(u_t^\top p)$ is $\alpha$-exp-concave for some $\alpha > 0$ for all $t \in [T]$, meaning that

$$\nabla^2 f_t(p) - \alpha \nabla f_t(p) \nabla f_t(p)^\top \succeq 0.$$

Let $A_t \subseteq [N]$ be the set of agents with $u_{t,:,n} = c_{t,n} \cdot \mathbf{1}$ for all $n \in A$ on round $t \in [T]$. It is guaranteed that $|A_t| \geq M$ for all $t \in [T]$. Denote $B_t = [N] \setminus A_t$. Direct calculation shows that

$$\nabla f_t(p) = \frac{\Pi_{m \in A_t} c_{t,m}^{\frac{1}{N}}}{N} \sum_{n \in B_t} \frac{f_t(p)}{\langle p, u_{t,:,n} \rangle} u_{t,:,n},$$

$$\nabla^2 f_t(p) = \frac{\Pi_{m \in A_t} c_{t,m}^{\frac{1}{N}}}{N} \sum_{i \in B_t} \frac{u_{t,:,n} \nabla f_t(p)^\top \langle p, u_{t,:,n} \rangle - f_t(p) u_{t,:,n} u_{t,:,n}^\top}{\langle p, u_{t,:,n} \rangle^2}$$

$$= \frac{f_t(p) \Pi_{m \in A_t} c_{t,m}^{\frac{2}{N}}}{N^2} \left( \sum_{i \in B_t} \frac{u_{t,:,n}}{\langle p, u_{t,:,n} \rangle} \right) \left( \sum_{n \in B_t} \frac{u_{t,:,n}}{\langle p, u_{t,:,n} \rangle} \right)^\top$$

$$- \frac{f_t(p) \Pi_{m \in A_t} c_{t,m}^{\frac{1}{N}}}{N} \sum_{i \in B_t} \frac{u_{t,:,n} u_{t,:,n}^\top}{\langle p, u_{t,:,n} \rangle^2}.$$

For notational convenience, let $\lambda_t = \Pi_{m \in A_t} c_{t,m}^{\frac{1}{N}} \leq 1$. Picking $\alpha = \frac{M}{N-M}$, we know that

$$\nabla^2 f_t(p) - \alpha \nabla f_t(p) \nabla f_t(p)^\top$$

$$= \frac{\lambda_t^2 f_t(p)}{N^2} \left( \sum_{n \in B_t} \frac{u_{t,:,n}}{\langle p, u_{t,:,n} \rangle} \right) \left( \sum_{n \in B_t} \frac{u_{t,:,n}}{\langle p, u_{t,:,n} \rangle} \right)^\top - \frac{\lambda_t f_t(p)}{N} \sum_{i \in B_t} \frac{u_{t,:,n} u_{t,:,n}^\top}{\langle p, u_{t,:,n} \rangle^2}$$

$$- \frac{\alpha \lambda_t^2 f_t(p)^2}{N^2} \left( \sum_{n \in B_t} \frac{u_{t,:,n}}{\langle p, u_{t,:,n} \rangle} \right) \left( \sum_{n \in B_t} \frac{u_{t,:,n}}{\langle p, u_{t,:,n} \rangle} \right)^\top$$

$$= \frac{-\lambda_t f_t(p)}{N} \left[ \sum_{n \in B_t} \frac{u_{t,:,n} u_{t,:,n}^\top}{\langle p, u_{t,:,n} \rangle^2} - \frac{\lambda_t (1 - \alpha f_t(p))}{N} \left( \sum_{n \in B_t} \frac{u_{t,:,n}}{\langle p, u_{t,:,n} \rangle} \right) \left( \sum_{n \in B_t} \frac{u_{t,:,n}}{\langle p, u_{t,:,n} \rangle} \right)^\top \right]$$

$$\succeq \frac{-\lambda_t f_t(p)}{N} \left[ \sum_{n \in B_t} \frac{u_{t,:,n} u_{t,:,n}^\top}{\langle p, u_{t,:,n} \rangle^2} - \frac{1 - \alpha f_t(p)}{N} \left( \sum_{n \in B_t} \frac{u_{t,:,n}}{\langle p, u_{t,:,n} \rangle} \right) \left( \sum_{n \in B_t} \frac{u_{t,:,n}}{\langle p, u_{t,:,n} \rangle} \right)^\top \right]$$

$$\succeq \frac{-\lambda_t f_t(p)}{N} \left[ \sum_{n \in B_t} \frac{u_{t,:,n} u_{t,:,n}^\top}{\langle p, u_{t,:,n} \rangle^2} - \frac{1 + \alpha}{N} \left( \sum_{n \in B_t} \frac{u_{t,:,n}}{\langle p, u_{t,:,n} \rangle} \right) \left( \sum_{n \in B_t} \frac{u_{t,:,n}}{\langle p, u_{t,:,n} \rangle} \right)^\top \right]$$

$$= \frac{-\lambda_t f_t(p)}{N} \left[ \sum_{n \in B_t} \frac{u_{t,:,n} u_{t,:,n}^\top}{\langle p, u_{t,:,n} \rangle^2} - \frac{1}{N-M} \left( \sum_{n \in B_t} \frac{u_{t,:,n}}{\langle p, u_{t,:,n} \rangle} \right) \left( \sum_{n \in B_t} \frac{u_{t,:,n}}{\langle p, u_{t,:,n} \rangle} \right)^\top \right]$$

$$\succeq \frac{-\lambda_t f_t(p)}{N} \left[ \sum_{n \in B_t} \frac{u_{t,:,n} u_{t,:,n}^\top}{\langle p, u_{t,:,n} \rangle^2} - \frac{1}{|B_t|} \left( \sum_{n \in B_t} \frac{u_{t,:,n}}{\langle p, u_{t,:,n} \rangle} \right) \left( \sum_{n \in B_t} \frac{u_{t,:,n}}{\langle p, u_{t,:,n} \rangle} \right)^\top \right]$$

$$\succeq 0, \qquad \qquad \text{(Cauchy-Schwarz inequality)}$$

where the first inequality is because $\lambda_t \leq 1$ and $f_t(p) \leq 0$; the second inequality is because $f_t(p) \geq -1$; the third inequality is because $|B_t| \leq \frac{1}{N-M}$. This shows that the $f_t(p)$ is $\frac{M}{(N-M)}$-exp-concave. Therefore, according to Theorem 4.4 of [Hazan et al., 2016], we know that the EWOO algorithm guarantees that

$$\sum_{t=1}^{T} (f_t(p_t) - f_t(p^\star)) \leq \left( \frac{N-M}{M} \right) \cdot K \log T + \frac{2(N-M)}{M}.$$

$\square$

# C   Auxiliary Lemmas

In this section, we include several auxiliary lemmas that are useful in the analysis.

**Lemma C.1.** *Let $a_1, \ldots, a_n, b_1, \ldots b_n \in [0, 1]$, where $a_i \geq b_i$ for all $i \in [n]$. Then, $\prod_{i=1}^n a_i - \prod_{i=1}^n b_i \leq \sum_{i=1}^n (a_i - b_i)$.*

*Proof.* Direct calculation shows that

$$
\prod_{i=1}^n a_i - \prod_{i=1}^n b_i = \sum_{j=1}^n \left( \prod_{i=1}^j a_i \prod_{i=j+1}^n b_i - \prod_{i=1}^{j-1} a_i \prod_{i=j}^n b_i \right)
$$
$$
= \sum_{j=1}^n \left( (a_j - b_j) \prod_{i=1}^{j-1} a_i \prod_{i=j+1}^n b_i \right)
$$
$$
\leq \sum_{j=1}^n (a_j - b_j).
$$

$\square$

**Lemma C.2.** *For all $x \in (0, 1)$ and $\alpha \in (0, 1)$ satisfying $1 + x \log \alpha \geq 0$, we have $\frac{1 - \alpha^x}{x} \geq -\frac{\log \alpha}{2}$.*

*Proof.* Let $y = \log \alpha < 0$. We know that

$$
\frac{1 - \alpha^x}{x} \geq -\frac{\log \alpha}{2}
$$
$$
\Longleftrightarrow 1 - e^{xy} \geq -\frac{xy}{2}
$$
$$
\Longleftrightarrow e^{xy} \leq 1 + \frac{xy}{2},
$$

which is true since $e^u \leq 1 + \frac{u}{2}$ for all $u \in [-1, 0]$ and $xy = x \log \alpha \geq -1$. $\square$

**Lemma C.3** (Theorem 1 in [Beygelzimer et al., 2011]). *Let $X_1, \ldots, X_T \in [-B, B]$ for some $B > 0$ be a martingale difference sequence and with $\sum_{t=1}^T \mathbb{E}_t[X_t^2] \leq V$ for some fixed quantity $V > 0$. We have for all $\delta \in (0, 1)$, with probability at least $1 - \delta$,*

$$
\sum_{t=1}^T X_t \leq \min_{\lambda \in [0, 1/B]} \left( \lambda V + \frac{\log(1/\delta)}{\lambda} \right) \leq 2\sqrt{V \log(1/\delta)} + B \log(1/\delta).
$$

