# OpenReview forum: "No-Regret Learning for Fair Multi-Agent Social Welfare Optimization"
_NeurIPS.cc/2024/Conference — NeurIPS 2024 poster_

### Official Review · Reviewer_izKh · 2024-06-16

**Soundness:** 2
**Presentation:** 3
**Contribution:** 2
**Rating:** 4
**Confidence:** 4

**Summary:**

The paper considers a new fairness measure, say NSW, lacking Lipschitzness. Unlike previous measures, the multi-armed bandit problem cannot obtain common $O(\sqrt T)$ regret. For stochastic MAB, the authors find an algorithm achieving $\tilde{O}(T^{\frac{N-1}{N}})$ regret and show its tightness. For adversary scenarios, the authors show that no algorithm can achieve sublinear regret. Then, they consider an easier information structure, say full-information feedback and $\sqrt T$ regret is possible. Finally, they consider the situation when logarithmic regret is achievable.

**Strengths:**

The model is clear and the algorithms proposed are easy to understand. Besides, the authors consider different information structures and models to depict the problem completely. And the construction of lower bounds is very nice.

**Weaknesses:**

1. The paper shows why considering $NSW$ is better than considering $NSW_{prod}$. However, why not consider simple average which is more intuitive? I recommend that the authors provide some realistic applications for NSW.

2. If I understand correctly, you consider an addictive utility, for example, you consider expected regret when showing lower bounds. However, for (2), why do you use $NSW(u^Tp_t)$ rather than $\sum_i p_{t,i}NSW(u_{t,i})$? The definition of regret seems strange.

3. The statement in Line 165 is not correct. We don't need to observe $u^T_t p_t$ when using BCO though we may need the dimension of $u$ to be 1, i.e. $N=1$. When you prove the lower bounds, I think the negative result, i.e., Thm 3.2, holds because NSW is concave but https://arxiv.org/pdf/2402.06535 needs convexity. However, in Appendix B. 1., you have convexity, so you can only construct a hard-to-learn example with at least two agents. I'm not sure whether my intuition is correct. If it is, some statement in the paper doesn't hold. If not, please explain more about the construction of your lower bounds to make it easy to understand.

4. You mention that NSW is not Lipschitz, but why not make a truncation? If an agent has a very small utility, like $\sigma$ in the paper, the regret will be small. Otherwise, we can use Lipschitzness. Is there any difference between your methods? If not, show the reason why NSW doesn't make the problem harder. Also, if you have time, it's meaningful to use experiments to compare these two ideas.

**Questions:**

1. What will happen if the learning can only observe NSW rather than everyone's utility? Will it change the dependence of $K$ and $N$?

2. In Line 324, what is $f(p_t)$? I believe the input of $f$ is an $R^N$ vector. However, $p_t$ belongs to $R^K$.

3. Typo: In Line 338, "is not only convex" should be "concave".

**Limitations:**

No limitations.

---

> ### Author Rebuttal · Authors · 2024-08-06
>
> We thank the reviewer for the detailed and valuable comments to our paper. We address the reviewer's questions as follows.
>
> **Q1: However, why not consider a simple average, which is more intuitive? I recommend that the authors provide some realistic applications for NSW.**
>
> A: It is well-known that a simple average does not promote fairness. To see this, consider the following example mentioned by Hossain et al. [2021]: suppose there are 10 agents and 2 arms, where the first 4 agents receive reward 1 from arm A and reward 0 from arm B, and the other 6 agents receive reward 0 from arm A and reward 1 from arm B. If we use average as the measure, then the best policy is to always pull arm B, while if we use NSW as the measure, the distribution (0.4, 0.6) over the two arms is the best policy, which is clearly more reasonable from a fairness viewpoint.
>
> For “some realistic applications for NSW”, we refer the reviewer to our first response to Reviewer t2CF.
>
> **Q2: For (2), why do you use $𝑁𝑆𝑊(u^\top p_t)$ rather than $\sum_{i\in[N]}p_{t,i}𝑁𝑆𝑊(𝑢_{𝑡,𝑖})$? The definition of regret seems strange.**
>
> A: This is because the former (geometric average of expected utilities) is arguably more meaningful as a fairness measure than the latter (expected geometric average of utilities). To see this, simply consider a setting with 2 agents and 2 arms, where the first agent always gets reward 1 from arm A and reward 0 from arm B, while the second agent is the opposite (reward 0 from arm A and reward 1 from arm B). Then, in terms of geometric average of expected utilities, the uniform distribution is the best policy (which makes sense from a fairness viewpoint); on the other hand, in terms of the expected geometric average of utilities, all distributions achieve the same value of 0, implying that all polices are as fair, which is clearly not what we want.
>
>
> **Q3: The statement in Line 165 is not correct. We don't need to observe 𝑢𝑡𝑇𝑝𝑡  when using BCO though we may need the dimension of 𝑢 to be 1, i.e. 𝑁=1. When you prove the lower bounds, I think the negative result, i.e., Thm 3.2, holds because NSW is concave but https://arxiv.org/pdf/2402.06535 needs convexity. However, in Appendix B. 1., you have convexity, so you can only construct a hard-to-learn example with at least two agents. I'm not sure whether my intuition is correct. If it is, some statement in the paper doesn't hold. If not, please explain more about the construction of your lower bounds to make it easy to understand.**
>
> A: We believe that there are some significant misunderstandings from the reviewer. First of all, when $N=1$, our problem simply reduces to the classic multi-armed bandit problem, whose minimax regret rate is well-studied, so our focus is on the case when $N\geq 2$. After all, what is the point of considering fairness when there is only one agent?
>
> Second, you mentioned that our NSW is concave while standard BCO needs convexity, but note that our problem is about **utility maximization** while BCO is about **loss minimization**, so concavity in the former is just equivalent to convexity in the latter by simply taking negation of the objective.
>
> Finally, your comment about Appendix B.1 also does not make sense to us. In Appendix B.1, we consider the product version of NSW used by previous works, which is neither convex nor concave.
>
> **Q4: You mention that NSW is not Lipschitz, but why not make a truncation? If an agent has a very small utility, like 𝜎 in the paper, the regret will be small. Otherwise, we can use Lipschitzness. Is there any difference between your methods? If not, show the reason why NSW doesn't make the problem harder. Also, if you have time, it's meaningful to use experiments to compare these two ideas.**
>
> A: We are not sure if we fully understand the reviewer’s idea, and will clarify based on our understanding below. If this does not address your question, please do follow up on this.
>
> First, note that our truncation in the analysis is on the quantity $\langle p_t, u_{:,n}\rangle$ (Line 202), which is related to the unknown utility $u$ and is thus cannot be explicitly implemented in the algorithm itself.
>
> Second, if what the reviewer meant is to truncate $\langle p_t, \hat{u}_{t,:,n}\rangle$ instead, that is, replace NSW in Algorithm 1 by
>
> $$NSW_{\sigma}(\mu)=\prod_{i=1}^N \max(\sigma, \mu)^{1/N},$$
>
> then, regardless whether the analysis works, it does not seem that the algorithm can be efficiently implemented since it is unclear how to compute $p_t$ given that $NSW_{\sigma}$ is not concave (and neither is $\log NSW_{\sigma}$).
>
> **Q5: What will happen if the learning can only observe NSW rather than everyone's utility? Will it change the dependence of $𝐾$ and $𝑁$ ?**
>
> A: As we discussed in Lines 161-171, in this case, the problem is just an instance of standard bandit convex minimization (or equivalently, bandit concave maximization). Therefore, applying the state-of-the-art algorithms from [Fokkema et al., 2024] achieves $O(K^{1.75}\sqrt{T})$ regret in the stochastic setting and $O(K^{3.5}\sqrt{T})$ in the adversarial setting (both without $N$ dependence).
>
>
> **Q6: In Line 324, what is $𝑓(𝑝_𝑡)$? I believe the input of $𝑓$ is an $\mathbb{R}^𝑁$ vector. However, $𝑝_𝑡$ belongs to $\mathbb{R}^𝐾$.**'
>
> A: It should be $f(u_t^\top p_t)$ instead. Thanks for pointing out the typo!
>
> **Q7: In Line 338, "is not only convex" should be "concave".**
>
> A: This is **not** a typo. Note that we are talking about -NSW here, which is indeed convex since NSW is concave.
>
> **Reference**
>
> [Fokkema et al., 2024] H. Fokkema, D. van der Hoeven, T. Lattimore, and J. Mayo. Online Newton method for bandit convex optimisation. In Conference on Learning Theory, 2024.

---

> > ### Comment · Reviewer_izKh · 2024-08-09
> >
> > Thank you for your response.

---

> > > ### Author Response · Authors · 2024-08-09
> > >
> > > Thanks for reading our response. If it addresses your concerns, please consider reevaluating our paper. Otherwise, please follow up with more questions. Thank you again.

---

> > > > ### Comment · Reviewer_izKh · 2024-08-13
> > > >
> > > > I will maintain my score for now and discuss it further with the other reviewers in the next round. My main concern remains with the definition of regret. You first use the geometric average $f(u^Tp)$ as a form of expectation, but then use the arithmetic mean to define regret (noting that you include an $\mathbb{E}$) as another kind of expectation. This approach does not seem entirely natural to me. I have no further questions.

---

### Official Review · Reviewer_aYwg · 2024-06-24

**Soundness:** 4
**Presentation:** 4
**Contribution:** 3
**Rating:** 7
**Confidence:** 3

**Summary:**

The authors address the problem of maximizing Nash Social Welfare (NSW) in a centralized multi-agent multi-armed bandit setting. In each round, a central entity selects a probability distribution $p_t$ over the $K$ actions, and draw an action $i_t  \sim p_t$.  Upon selecting an action, the central entity observes the $i$-th row of the $K\times N$ utility matrix $u_t$ which indicates the utility received by each of the $N$ agent when choosing one of the $K$ actions.The goal of the central entity is to maximize the sum of the Nash Social Welfares over the rounds, where the NSW for a given round is defined as the geometric mean of the expected utilities received by the agents: $$NSW(p_t, u_t) = \Pi_{j = 1}^{N} \langle p_t \vert u_t\rangle^{1/N}.$$

The authors note that, unlike the product of expected utilities, the Nash Social Welfare (NSW) is not Lipschitz continuous. Consequently, the analytical techniques used in previous works that study the product of expected utilities are not applicable here. In the scenario where the utility matrix $u_t$is drawn i.i.d. from some unknown distribution, the authors develop an algorithm for this problem and establish both upper and lower bounds that match in terms of scaling with the horizon $T$. Next, they consider the adversarial framework. They demonstrate that with bandit feedback, the worst-case regret must scale linearly. They propose two different algorithms for the adversarial case with full information and show that both achieve a regret of $\tilde{O(\sqrt{T}}$, with different dependence in $N$ and $K$.  Finally, they show that in some special cases, the regret can be logarithmic in $T$.

**Strengths:**

I am not very familiar with the literature on Nash Social Welfare; however this paper seem to address a well-motivated problem in multi-agent decision-making.

The authors propose a UCB algorithm for maximizing the sum of Nash Social Welfares (NSW), utilizing new confidence bounds to achieve optimal regret rates. They establish a matching lower bound on the regret, providing theoretical robustness to their approach. Additionally, they explore the adversarial case, a scenario that had not been previously addressed. Their results reveal that bandit feedback alone is insufficient to achieve sub-linear regret rates. To complete their investigation, they propose a Follow-the-Regularized-Leader (FTRL) algorithm. While FTRL is not a novel algorithm, its application here concludes a comprehensive analysis of the problem at hand.

The paper is very clear, with the key ideas of the proofs well outlined and the problem effectively contrasted with existing works.

**Weaknesses:**

I see no major weaknesses. As a minor remark, the use of the Nash Social Welfare is not motivated. While it is a standard welfare measure in economics, it may be unfamiliar to those in the bandit literature. Therefore, it is worth introducing and discussing its significance and relevance.

**Questions:**

- Could you please quickly comment on why the Nash Social Welfare is defined as the geometric average of the expected utility, instead of the expectation of the geometric averages of the utilities? Admittedly, the latter might be less interesting from a mathematical standpoint. Is this a choice you are making, or is it a commonly accepted definition?
- Could you please detail a bit more how you obtain the thrid line from the second line of the computations at the end of page 14?
- There is a typo at the last line of page 4, where a "O" is capitalized in the middle of a sentence.

**Limitations:**

The limitations are well addressed.

---

> ### Author Rebuttal · Authors · 2024-08-06
>
> Thanks for the reviewer's detailed and valuable comments to our paper. We address the reviewer's questions as follows.
>
> **Q1: the use of the Nash Social Welfare is not motivated**
>
> A: We will add more discussion on the significance and relevance of the use of NSW in the next version as suggested. We point out that Hossain et al. [2021], the main work we improve upon, has provided very good motivation on studying NSW, and we copy one such discussion below for reference:
>
> *This problem of making a fair collective decision when the available alternatives affect multiple agents is well-studied in computational social choice. The literature offers a compelling fairness notion called the Nash social welfare, named after John Nash… Maximizing the Nash social welfare is often seen as a middle ground between maximizing the utilitarian social welfare (sum of utilities to the agents), which is unfair to minorities, and maximizing the egalitarian social welfare (minimum utility to any agent), which is considered too extreme. The solution maximizing the Nash social welfare is also known to satisfy other qualitative fairness desiderata across a wide variety of settings.*
>
> **Q2: Could you please quickly comment on why the Nash Social Welfare is defined as the geometric average of the expected utility, instead of the expectation of the geometric averages of the utilities? Admittedly, the latter might be less interesting from a mathematical standpoint. Is this a choice you are making, or is it a commonly accepted definition?**
>
> A: We point out that the former (geometric average of expected utilities, which we study) is arguably more meaningful as a fairness measure than the latter (expected geometric average of utilities). To see this, simply consider a setting with 2 agents and 2 arms, where the first agent always gets reward 1 from arm A and reward 0 from arm B, while the second agent is the opposite (reward 0 from arm A and reward 1 from arm B). Then, in terms of geometric average of expected utilities, the uniform distribution is the best policy (which makes sense from a fairness viewpoint); on the other hand, in terms of the expected geometric average of utilities, all distributions achieve the same value of 0, implying that all polices are as fair, which is clearly not what we want.
>
> Therefore, both previous works (Hossain et al. [2021] and Jones et al. [2023]) and ours study the former notion (which is in fact also mathematically more interesting as the reviewer pointed out).
>
> **Q3: Could you please detail a bit more how you obtain the third line from the second line of the computations at the end of page 14?**
>
> A: We use the fact that $a^N-b^N=(a-b)\cdot (\sum_{k=0}^{N-1} a^kb^{N-1-k})$ with $a=<p_t, \hat{u}_{t,:,n}>^{1/N}$ and $b=<p_t, u_{:,n}>^{1/N}$.
>
> Finally, thanks for pointing out the typo on Page 4! We will fix that in the next revision.

---

> > ### Comment · Reviewer_aYwg · 2024-08-09
> > **Rebuttal acknowledgment**
> >
> > I thank the authors for their rebuttal, which adequately answers my questions.

---

### Official Review · Reviewer_t2CF · 2024-07-11

**Soundness:** 3
**Presentation:** 3
**Contribution:** 3
**Rating:** 6
**Confidence:** 3

**Summary:**

The paper studies the problem of online social welfare maximization. More precisely, the authors consider the online learning setting where the learner, at each round $t \in [T]$, picks an action $i \in [K]$ that then determines the utility of each of the $n$ agents. The utility of agent $j$ is given by the $(i,j)$ entry of a utility matrix $u_t \in \mathbb{R}^{K \times n}$, which can either change arbitrarily or be sampled from a predetermined probability distribution. The cost of the learner is given by the Nash Welfare, defined as the geometric mean of the agents' utilities: $NW(\mu) = (\Pi_{n \in [n] \mu_n})^{1/n}$. The goal of the learner is to pick a sequence of mixed actions to achieve social welfare comparable to that achieved by the best fixed mixed actions.

The authors consider both the stochastic and the deterministic versions of the problem, as well as both the bandit and full-information feedback models. For the stochastic case with bandit feedback (where the learner learns only the utilities of the agents for the randomly sampled action), the authors provide an $O(T^{N-1/N})$-regret online learning algorithm and show that this regret bound is tight for the stochastic setting. They then study the bandit feedback model with adversarial changes and establish an $\Omega(T)$ regret lower bound. In view of this negative result, the authors shift their attention to the full-information feedback and adversarial changes case, for which they provide $O(\sqrt{T})$-regret online learning algorithms using Follow the Regularized Leader with log-barrier regularization.

**Strengths:**

The paper studies an interesting and technically challenging setting. The writing is relatively good, and the authors clearly explain their contributions and the crucial differences between their setting and classical online concave optimization with bandit feedback. I found it particularly interesting that there is a significant discrepancy in the possible regret bounds between online concave optimization and the bandit feedback case of the considered setting. Additionally, I appreciate that the authors present both upper and lower bounds for all the considered settings.

**Weaknesses:**

Despite the paper's solid technical contribution, my only concern lies with the motivation for the setting. The authors briefly mention that the setting has applications in resource allocation but do not provide any concrete examples or a convincing discussion on why this setting is particularly interesting. While I do not doubt that the setting is indeed interesting, I believe a detailed discussion on the potential applications of the model would significantly enhance the paper.

**Questions:**

Could you elaborate more on the potential applications of the considered setting?

**Limitations:**

Yes

---

> ### Author Rebuttal · Authors · 2024-08-06
>
> Thanks for the reviewer's appreciation of our paper! We answer the reviewer's question as follows.
>
> **Q1: A detailed discussion on the potential applications of the model.**
>
> One important application of the model is fair repeated policy decision making. In fact, we believe that Hossain et al. [2021], the main work we improve upon, already provided very good justification for studying this model, and we copy one such discussion from their introduction below:
>
> *This problem can model situations where the principal is deliberating a policy decision and the arms correspond to the different alternatives she can implement. However, in many real-life scenarios, making a policy decision affects not one, but several agents. For example, imagine a company making a decision that affects all its employees, or a conference deciding the structure of its review process, which affects various research communities…This problem of making a fair collective decision when the available alternatives affect multiple agents is well-studied in computational social choice. The literature offers a compelling fairness notion called the Nash social welfare, named after John Nash… Maximizing the Nash social welfare is often seen as a middle ground between maximizing the utilitarian social welfare (sum of utilities to the agents), which is unfair to minorities, and maximizing the egalitarian social welfare (minimum utility to any agent), which is considered too extreme. The solution maximizing the Nash social welfare is also known to satisfy other qualitative fairness desiderata across a wide variety of settings.*

---

> > ### Comment · Reviewer_t2CF · 2024-08-09
> > **Reviewer's Response**
> >
> > Thank you for the response. I maintain my positive opinion for the paper and I keep my score.

---

### Decision · Program_Chairs · 2024-09-25

**Decision:**

Accept (poster)

**Comment:**

This work derives minimax regret bounds for the objective of Nash social welfare (NSW). Prior work did consider a similar objective (product of utilities) but the NSW objective (i.e. geometric mean) presents some unique challenges and requires new proof techniques. I also like the adversarial setting considered in the paper, and the paper is well-written and all the ideas are carefully explained. In the future version of the paper, it would be useful to have some discussions about the applications of NSW.